# The Indonesian Throughflow Circulation Under Solar Geoengineering

Chencheng Shen[1] John C. Moore[1,2]* Heri Kuswanto[3,4] Liyun Zhao[1]*

[1]State Key Laboratory of Earth Surface Processes and Resource Ecology, Faculty of Geographical Science, Beijing Normal University, Beijing 100875, China

[2]Arctic Centre, University of Lapland, Rovaniemi, Finland

[3]Center for Disaster Mitigation and Climate Change, Institut Teknologi Sepuluh Nopember, Surabaya, Indonesia.

[4]Department of Statistics, Institut Teknologi Sepuluh Nopember, Surabaya, Indonesia.

*Correspondence to:* john.moore.bnu@gmail.com, zhaoliyun@bnu.edu.cn

Short summary (less than 500 characters):

The Indonesia Throughflow is an important pathway connecting the Pacific and Indian Oceans and is part of a wind-driven circulation that is expected to reduce under greenhouse gas forcing. Solar dimming and sulfate aerosol injection geoengineering may reverse this effect. But stratospheric sulfate aerosols affects winds more than simply "shading the sun" and hence reduces the water transport similar as we simulate for unabated greenhouse gas emissions.

**Abstract**

The Indonesia Throughflow (ITF) is the only low-latitude channel between the Pacific and Indian oceans, and its variability has important effects on global climate and biogeochemical cycles. Climate models consistently predict a decline in ITF transport under global warming, but it has not yet been examined under solar geoengineering scenarios. We use standard parameterized methods for estimating ITF: the Amended Island Rule and Buoyancy Forcing, to investigate ITF under the SSP2-4.5 and SSP5-8.5 greenhouse gas scenarios, and the geoengineering experiments G6solar and G6sulfur that reduce net global mean radiative forcing from SSP5-8.5 levels to SSP2-4.5 levels using solar dimming and sulfate aerosol injection strategies. Six model ensemble mean projections for 2080 - 2100 relative to historical

(1980-2014) ITF are reductions of 19% under the G6solar scenario and 28% under the G6sulfur scenario
which compare with reductions of 23% and 27% under SSP2-4.5 and SSP5-8.5. Despite standard
deviations amounting to 5-8% for each scenario, all scenarios are significantly different from each other
($p<0.05$) when taken over the whole 2020-2100 simulation period. Thus, significant weakening of the
ITF occurs under all scenarios, but G6solar closer approximates SSP2-4.5 than does G6sulfur. In contrast
with the other three scenarios which show only reductions in forcing due to ocean upwelling, the G6sulfur
experiment shows a large reduction in ocean surface wind stress forcing accounting for 47% (38% - 65%
across model range) of the decline of Wind+Upwelling ITF transport. There are also reductions in deep-
sea upwelling in extratropical western boundary currents.

**1. Introduction**
The Indonesian Throughflow (ITF) is an important part of the global thermohaline circulation (Gordon,
1986; Lee et al., 2002; Sprintall et al., 2009). The ITF brings about 15 Sv (1 Sv = $10^6 \, m^3/s$; ~10.7 to ~18.7
Sv during the INSTANT Field Program, 2004-2006) of warm and fresh water from the Pacific to the
Indian Ocean (Sprintall et al., 2009). Since the ITF is the only ocean pathway in the tropics between the
Pacific and Indian Oceans it is the key to heat and water volume transport between them (Godfrey, 1996;
Talley, 2008). The ITF also plays an important role in regulating global climate and biogeochemical
cycles (Ayers et al., 2014; Hirst and Godfrey, 1994), for example  the ITF may influence the El Nino-
Southern Oscillation (ENSO) by altering the tropical–subtropical exchange, the structure of the mean
tropical thermocline, and the mean sea surface temperature (SST) difference between the Pacific warm
Pool and the cold tongue, etc. (Lee et al., 2002) and in the supply of iron in the equatorial upwelling,
maintaining biological production in the equatorial eastern Pacific (Gorgues et al., 2007). Sen Gupta et
al. (2021) used 26 CMIP6 models to predict ITF weakening by 3 Sv (2.4-3.2 Sv model range) under the
SSP5-8.5 scenario (the high greenhouse gas emission scenario) relative to 20[th] century historical means
The decline in the ITF would lead to more heat to accumulate in the Pacific Ocean, which could alter
tropical atmospheric-ocean interactions and contribute to extreme El Nino /La Nina events (Cai et al.,
2015; Klinger and Garuba, 2016).

The ITF is fed by the Mindanao Current and the New Guinea Coast Undercurrent  (Figure 1) and, to a
lesser extent, parts of the low-latitude Pacific Western Boundary Current (WBC) that flows toward the
equator (Godfrey, 1996; Lukas et al., 1996). The ITF helps supply the Agulhas current leakage from the
Indian Ocean to the South Atlantic Ocean, and may be said to flush Indian Ocean thermocline waters
southward by boosting the Agulhas current (Durgadoo et al., 2017; Gordon, 2005).

The interannual and decadal variability of the ITF transport is influenced by surface winds in the Pacific
and Indian Oceans (Feng et al., 2011; Meyers, 1996). Wyrtki (1987) noticed that the pressure gradient
between the Pacific and Indian Oceans dominates the ITF flux, and hence that sea level is a good indicator
of upper-ocean ITF transport. The largest volume flux is in July-August and the lowest in January-
February.

Model simulations consistently project that ITF transport will be weakened by increased greenhouse gas
(GHG) forcing (Feng et al., 2012; Hu et al., 2015; Sen Gupta et al., 2021; Vecchi and Soden, 2007). The
driving force is the weakening of the Pacific trade winds under global warming in the 21$^{st}$ century which
then weaken the Mindanao Current, the main inflow route of the ITF (Alory et al., 2007; Duan et al.,
2017; Sen Gupta et al., 2012).

### a) The wind stress integral path and buoyancy region

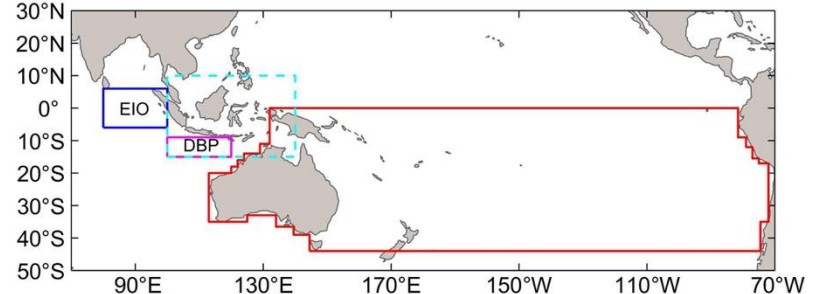

### b) Topography of Indonesian Sea

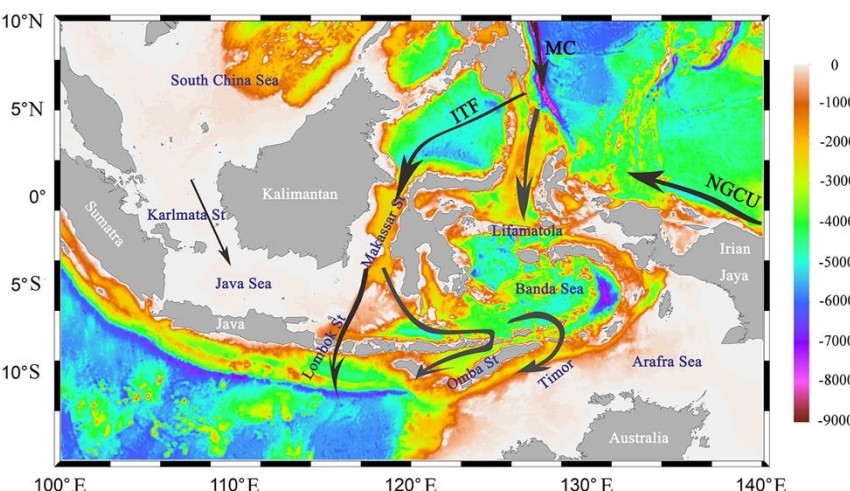

Figure 1. (a) The red line is the wind stress integral path for the Island Rule, The Downstream Buoyant Pool (magenta box) and Equatorial Indian Ocean (blue box) where the density difference is the main index to calculate the ITF transport by buoyance forcing. (b) Inset defined by the cyan dotted line in the panel (a) showing the offshore bathymetry in the maritime continent (ETOPO Global Relief Model, (Amante and Eakins, 2009)) and the Mindanao Current (MC), and the New Guinea Coast Undercurrent (NGCU) paths contributing to the ITF.

Analyzing the water flux through the many shallow channels in the Indonesian archipelago is challenging, and many of these channels are not resolved in simulations with resolutions of a degree or so (Gordon et al., 1999) (Figure 1). This motivates use of alternative methods of estimating ITF. Godfrey (1989) created the Island Rule to estimate flux based on Sverdrup theory (Sverdrup, 1947) analysis of Pacific wind stress. More recently, analysis of climate models revealed the importance of deep ocean circulation to the reduction of ITF transport under GHG forcing. The decline in ITF under GHG forcing could be due

to both the weakening of trade winds in the Pacific, and deep ocean circulation changes (Feng et al.,
2012; Hu et al., 2015). Interannual to decadal, as well as centennial dependence of the ITF on wind and
upwelling was found with an eddy-resolving ocean model simulation (Feng et al., 2017). This led to Sen
Gupta et al. (2016), and Feng et al. (2017) proposing the Amended Island Rule that modifies the Island
Rule to include the estimated net Pacific upwelling contribution to ITF based on high-resolution (1/10°)
ocean general circulation modelling.

An alternative mechanism for the ITF driver was proposed earlier by Andersson and Stigebrandt (2005).
In this theory buoyancy forcing is more important than wind forcing in driving the ITF. The ITF
variability is found from the baroclinic outflow of the Downstream Buoyant Pool (DBP) that extends
over much of the North Australian Basin (Figure 1). Hu and Sprintall (2016) used this method with
reanalysis products to produce ITF interannual variability in good agreement with the observed volume
transports (2004–2006) from the INSTANT mooring array transport (Sprintall et al., 2009), although the
average transport was only half the transport observed during INSTANT. INSTANT uses moorings
deployed at the major inflow (Makassar Strait, Lifamatola Strait) and outflow passages (Lombok Strait,
Ombai Strait and Timor Passage) of the ITF to estimate the ITF transport, resulting in a value of 15 Sv
during 2004-2006.  In contrast with the reasonable agreement for the Amended Island Rule estimates of
ITF, the alternative buoyancy method behaves much worse, indicating that the hypothetical forcing is
not as good an explanation for ITF as the Amended Island Rule, or that the models used do not capture
the specific details of the DBP. But although the Amended Island Rule matches the short duration of
observed fluxes and variability better than buoyancy, it is possible that changes in buoyancy forcing may
affect volume transport of the ITF on decadal scales under a changing climate and so we examine its
changes under the geoengineering scenarios.

Solar Radiation Modification (SRM) geoengineering is designed to reduce the solar radiation reaching
the surface of the earth and slow down climate warming due to GHG forcing (Shepherd, 2009). Since
SRM shortwave forcing has different spatial and temporal variability than longwave forcing, it can only
imperfectly offset the climate change caused by the increase of GHGs. In this article we focus on two

styles of SRM: reduction of the solar constant to mimic the effect of a sunshade, called solar dimming

(SD); and stratospheric aerosol injection (SAI), specifically with injection of sulfate aerosol in the

tropical lower stratosphere (Kravitz et al., 2015). These styles of SRM are known to produce over-cooled

tropical oceans and under-cooled poles relative to global mean temperatures. However, other styles of

injection strategies than the simple tropical site specified by G6 can produce simulated climates without

these temperature biases (MacMartin and Kravitz, 2016). Simulated tropical atmospheric circulation

systems are impacted under both GHG and solar geoengineering scenarios. Under SD, the seasonal

movement of the intertropical convergence zone is reduced relative to GHG climates (Smyth et al., 2017).

Both the Hadley and Walker circulations are different from the historical (Cheng et al., 2022; Guo et al.,

2018). Impacts of SRM on the Walker circulation are modest compared with the Hadley cell but appear

most obviously in relation to the South Pacific Convergence Zone (Guo et al., 2018), which is relevant

in the overall tropical Pacific atmosphere system that drives and interacts with the ITF. Greenhouse gas

forcing is expected to cause an expansion of the Hadley circulation cells which may be asymmetric

between northern and southern hemispheres (Staten et al., 2019). Both SD (Guo et al., 2018) and SAI

(Cheng et al., 2022) reduce these greenhouse gas induced changes in the Hadley circulation, although

again hemispheric differences remain, and in the Cheng et al. (2022) simulations, were associated with

stratospheric heating and tropospheric temperature response due to enhanced stratospheric aerosol

concentrations. The changes in stratospheric heating, the tropopause height, and tropical sea surface

temperatures may be expected to impact tropical cyclogenesis, and this is consistent with reduction in

North Atlantic hurricane numbers and intensity relative to GHG-only climates under SAI (Moore et al.,

2015). However, there are differences between tropical basins in expected tropical cyclogenesis potential

and significant differences in simulations between climate models (Wang et al., 2018). Potential energy

available for extratropical storms is also consistently reduced under SRM relative to GHG

forcing(Gertler et al., 2020). The reported impacts highlight the potential role of wind forcing in ITF.

Little research to date has been done on ocean circulation under SRM, with only the Atlantic Meridional

Overturning Circulation (AMOC) having been studied in depth (Hong et al., 2017; Moore et al., 2019;

Muri et al., 2018; Tilmes et al., 2020; Xie et al., 2022). Both GHG forcing alone, and with SRM, produce

a weakening of AMOC relative to present day, mainly in response to the change of heat flux in the North
Atlantic, with little influence from the changes of freshwater flux and wind stress (Hong et al., 2017; Xie
et al., 2022). AMOC is less weakened under SRM than with GHG forcing alone and the AMOC declines
seen under GHG forcing are consistently reversed by SRM towards present day patterns (Moore et al.,
2019; Muri et al., 2018; Tilmes et al., 2020).

In this study, we will examine the impact of SRM on the change of the ITF in the 21st century, and
consider the transport and drivers differences between pure GHG climates representing moderate
mitigation (SSP2-4.5) and no mitigation (SSP5-8.5); with solar dimming (G6solar) and stratospheric
aerosol injection (G6sulfur) forms of SRM geoengineering.

**2. Climate Models and Scenarios**
The Intergovernmental Panel on Climate Change (IPCC) Shared Socioeconomic Pathways (SSPs) are
scenarios defined by radiative forcing goals to be achieved through various climate mitigation policy
alternatives (Kriegler et al., 2012; van Vuuren et al., 2011). The climate model simulation results under
the SSPs are being performed as part of the Coupled Model Intercomparison Project Phase 6 (CMIP6).
We used CMIP6 historical simulation during 1980-2014 (Eyring et al., 2016) and two GHG scenarios
during 2015-2100: SSP5-8.5, an unmitigated GHG emission scenario which raises mean global radiative
forcing by 8.5 W/m$^2$ over pre-industrial levels at 2100; and SSP2-4.5 designed to reach peak radiative
forcing of 4.5 W/m$^2$ by mid-century (O'Neill et al., 2016). We use the Geoengineering Model
Intercomparison Project Phase 6 (GeoMIP6) G6sulfur and G6solar scenarios during 2020-2100 (Kravitz
et al., 2015). The G6sulfur experiment specifies using SAI to reduce the net anthropogenic radiative
forcing constantly during the 2020-2100 period from the SSP5-8.5 to the SSP2-4.5 level, while G6solar
does the same using SD (Kravitz et al., 2015). The two SRM methods produce significantly different
surface climates, with differences from SSP2-4.5 being larger and more spatially variable under G6sulfur
than G6solar (Visioni et al., 2021). While the G6 scenarios are not particular realistic, for example they
specify starting SAI in 2020 and specify a very simple tropical injection strategy, they do provide a
usefully large SRM and GHG signal, and have been simulated by six CMIP6 generation models. This
allows more robust findings of the general impacts of SAI, especially when considering aspects of the
climate system that have not been addressed to date in geoengineering studies, such as the ITF.

We used monthly data from the first realization in each scenario from all six Earth System Models (ESM;
Table 1) that have performed the CMIP6 and GeoMIP6 scenarios to estimate the ITF transport. The
variable fields we use are zonal and meridional wind stress (tauu and tauv), sea water vertical velocity
(wo), sea water salinity and temperature (so and thetao). All fields were bi-linearly interpolated (except
for sea water vertical velocity, for which we use conservative interpolation) onto a common $0.5° \times 0.5°$
grid.

**Table 1**
*Earth System Models (ESMs) Used in This Study*

| Model | Atmospheric Resolution (long × lat) | Ocean Resolution (long × lat) | Reference |
|---|---|---|---|
| CESM2-WACCM | 288 × 192 | 320 × 384 | (Danabasoglu et al., 2020) |
| CNRM-ESM2-1 | 256 × 128 | 362 × 294 | (Séférian et al., 2019) |
| IPSL-CM6A-LR | 144 × 143 | 320 × 384 | (Boucher et al., 2020) |
| MPI-ESM1-2-HR | 384 × 192 | 802 × 404 | (Mauritsen et al., 2019) |
| MPI-ESM1-2-LR | 192 × 96 | 256 × 220 | (Mauritsen et al., 2019) |
| UKESM1-0-LL | 192 × 144 | 360 × 330 | (Sellar et al., 2019) |


**3. Methods**
**3.1 Island Rule**
In the Sverdrup balance, ocean current acceleration and friction are neglected, and wind stress curl is the
driving force of large-scale ocean circulation (Sverdrup, 1947). The "Island Rule" (Godfrey, 1989) uses
the Sverdrup balance to calculate the net total flow through a region by the integral of the wind stress on
a specific closed path. This  is a simple and more efficient way of estimating the long-term magnitude
and interannual variability than direct observations of flow through the complex channel topography and
equator spanning Indonesian archipelago (Godfrey, 1996). Feng et al. (2011) used an eddy‑permitting
numerical model, ORCA025, to verify that the Island Rule can capture the decadal variability of the ITF
transport.

The original Island Rule assumes that the ocean is dormant below a moderate depth, *Z*, below which
there is no motion (Sverdrup, 1947). The ITF transport is determined by the integral of wind stress along
the path from the southern tip of Australia, eastwards to South America, following the coastline to the
latitude line of the northwestern tip of Papua New Guinea (PNG) and then traces the west coast of
Australia back to the starting point (Figure 1a):
$$T_{ITF} = \frac{1}{f_N - f_S} \oint \frac{\tau^l}{\rho_0} dl \qquad (1)$$

where $f_N$ and $f_S$ are the Coriolis parameter at the equator and 44°S, respectively. $\tau^l$ is the along route
wind stress component. $\rho_0$ is the mean sea water density.

**3.2 Amended Island Rule**
Studies have suggested that a decline in ITF under GHG forcing was due to both the weakening of trade
winds in the Pacific, and the impact of the deep ocean circulation change (Feng et al., 2012; Hu et al.,
2015). Sen Gupta et al. (2016) used a climate model to attribute GHG-forced decrease of the ITF transport
to weakening of deep Pacific upwelling. Feng et al. (2017) estimated the contribution of deep ocean
upwelling from the Pacific north of 44°S to produce the Amended Island Rule:
$$T_{ITF} = \frac{1}{f_N - f_S} \oint \frac{\tau^l}{\rho_0} dl + \iint_{pacific} w_z \, ds \qquad (2)$$

where $w_z$ is the vertical velocity of the Pacific at 1500 m depth. The contribution of deep ocean upwelling
is integrated over the whole Pacific north of 44°S (considering volume conservation and the sill depths
of the Indonesian seas is less than 1500m). The Amended Island Rule was verified with a near-global
eddy-resolving ocean model simulation, and found to well-estimate the interannual to decadal, as well
as centennial variabilities of the ITF transport (Feng et al., 2017). Here we describe the ITF using the
Amended Island Rule, and its component parts which are the wind driven Sverdrup balance, and the
Pacific upwelling.

**3.3 Buoyancy Forcing**

Sea levels in the Pacific and Indian Oceans have been used to estimate the ITF transport in previous studies (Clarke and Liu, 1994; Potemra et al., 1997; Susanto and Song, 2015). Buoyancy accounts for high steric sea level (that is a volume increase due to lower density) in the North Pacific (Stigebrandt, 1984). A pool of low-density water (the DBP) originating in the North Pacific is formed in the eastern Indian Ocean between the Indonesian islands and northwestern Australia (Figure 1a). The sea level drop between Indian and Pacific Oceans occurs essentially at the abrupt eastern boundary of the DBP and is the source of buoyancy forcing (Andersson and Stigebrandt, 2005). In the DBP region, the long-term difference between the westward and eastward transport along the northern and southern flanks of the pool is the ITF transport.

The geostrophic transport in the DBP is related to denser water in the eastern equatorial Indian Ocean (EIO):

$$Q_\lambda = \frac{gH^2\Delta\rho}{2f_\lambda\rho_0} \tag{3}$$

$$ITF = Q_{\lambda_N} - Q_{\lambda_S} \tag{4}$$

where $g$ is acceleration due to gravity, $H$ is the penetration depth of the DBP (set by (Andersson and Stigebrandt, 2005) as 1200 m), $f_\lambda$ is the Coriolis parameter at latitude $\lambda$, $\rho_0$ is the reference density at 1200 m, The northern ($\lambda_N$) and southern ($\lambda_S$) boundary latitudes of the DBP are 10°S and 16°S respectively. $\Delta\rho$ is the density difference between the DBP region (9°S–15°S, 100°E–120°E) and the EIO region (6°N–6°S, 80°E–100°E). Hu and Sprintall (2016) verified the use of DPB and EIO to calculate $\Delta\rho$ with observations.

**4. Transport and Geoengineering**

**4.1 ITF Transport**

The multi-model ensemble mean wind driven ITF transport is ~16.9 Sv with the Pacific upwelling north of 44°S contributing ~4.5 Sv in the historical period (Figure 2). This compares with observational estimates of about 15 Sv during 2004-2006 (Sprintall et al., 2009) and the multi-model ensemble (total 22 CMIP5 models) mean is 15.2 Sv during 1900-2000 (Sen Gupta et al., 2016). Under SSP2-4.5 during

2015 - 2100, the wind-driven and Pacific upwelling contributions to ITF transport are not much different from those under SSP5-8.5. The wind driven volume ITF transport has significant trends for all scenarios with smallest trends for the SSP scenarios (linear trends of lower magnitude than 0.02 Sv per year), while the upwelling contributions has obvious downward trends in all scenarios. These trends appear to be consistent, despite differences in estimated transport across models (Figure S1). Thus the decline in future ITF transport in future GHG climates was explained by (Feng et al., 2017) as due to weakening of the Pacific upwelling on centennial timescales while wind-driven processes had no impact on long timescales.

During the last 20 years of the 21st century, the simulated ITF transport using the Amended Island Rule is 27% ± 3% (standard error) under SSP5-8.5 (Figure 2c), with Pacific upwelling decline accounting for 76%±15% (p<0.05, Wilcoxon signed rank test) of the Wind+Upwelling reduction. Both wind driven and upwelling contributions to ITF transport are slightly higher under SSP2-4.5 than under SSP5-8.5 during the same period, but the differences are small over the whole 2015-2100 period. The Wind+Upwelling ITF transport is reduced by 23%±2% (standard error, p<0.05) under SSP2-4.5 during the period of 2080-2100 relative to the historical period (13%~27% cross ESM range), and with the wind driven component only dropping by 5% (-2%~9% range). The reductions under SSP5-8.5 for upwelling and wind driven components are respectively 97% (60%~305%) and 8% (1%~19%).

In contrast with the reasonable agreement for the Amended Island Rule estimates of ITF, the alternative buoyancy method behaves much worse. The multi-mean ITF transport simulated by buoyancy forcing is 7.3 Sv in the historical period, which is less than that by wind driven and only half the transport observed during INSTANT (Sprintall et al., 2009), and there is large across-model variability (Figure S2). Under the two SSPs scenarios, the difference in ITF transport is small with significant trend during 2015-2100. The buoyancy driven estimation method can capture the interannual variability of ITF transport, but it does not perform well on centennial timescales (Hu and Sprintall, 2016), where ITF is much closer to that from the wind driven estimation method.

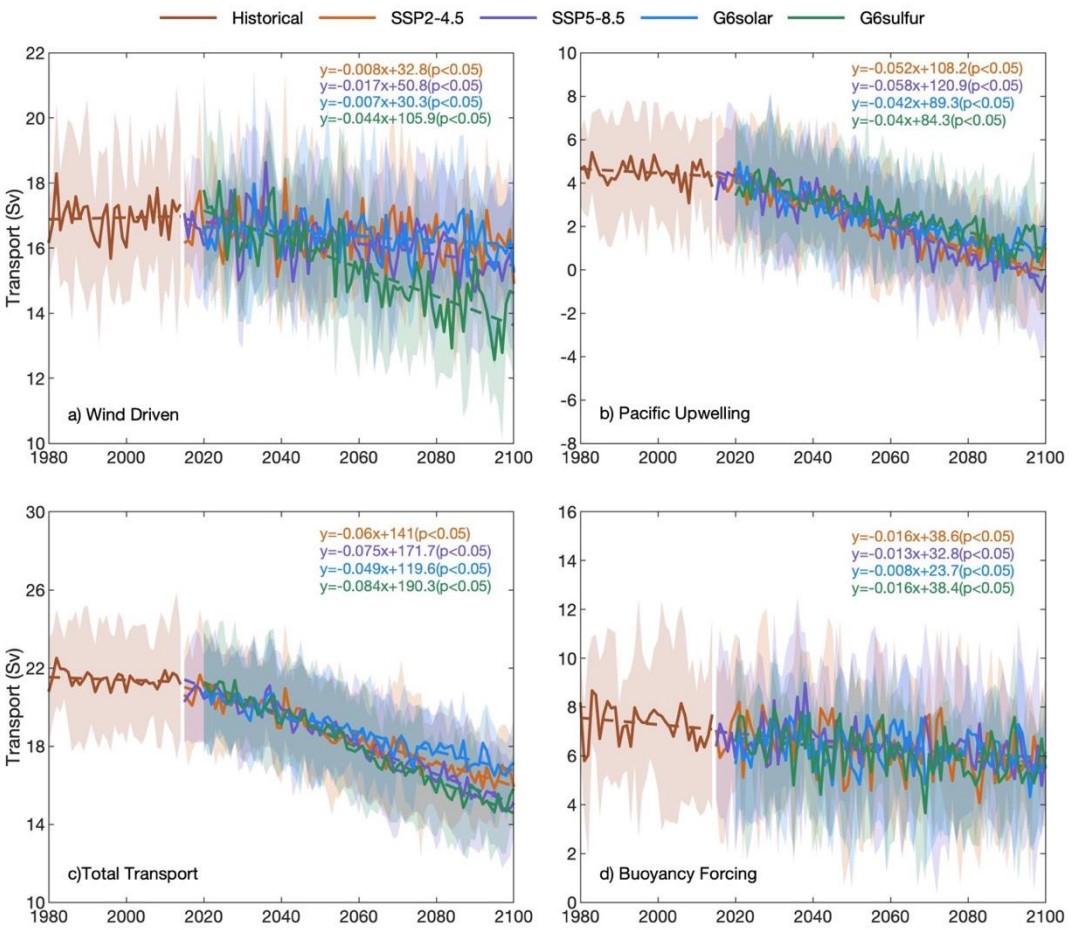

**Figure 2**. Six ESM ensemble mean ITF components under different scenarios, shadings show the standard deviation and the equation are the regression trend lines (2015-2100 under the two SSP scenarios and 2020-2100 under the two G6 scenarios) and the significance of the slope, (a) Sverdrup balance wind driven component. (b) Pacific upwelling north of 44°S. (c) Wind+Upwelling ITF under the Amended Island Rule (Eq 2). (d) ITF transport by buoyancy forcing. Individual ESM results are shown in Figure S1.

SAI and SD geoengineering methods clearly have different impacts on wind driven contributions to ITF transport for all models (Table S1) and the ensemble mean (Table 2) according to the Wilcoxon signed-rank test, and smaller although still significant differences in upwelling for the 6 model ensemble mean, although significant differences individually only for CESM2-WACCM (Figure 2a,b, Table 2; Table S1). Under the G6solar and G6sulfur scenarios, the Wind+Upwelling ITF transport is reduced by 19%±1% and 28%±1% respectively during 2080 - 2100 relative to the historical period, of which the wind-driven

ITF transport is reduced by 4%±1% and 16%,±1%, and the upwelling transport volume is reduced by
76%±8% and 70%±10%, all these differences between scenarios are significant (p<0.05, Wilcoxon
signed-rank test; Table 2). Under G6sulfur, the wind driven ITF transport has a clear downward trend in
contrast with the other three climate scenarios (Figure 2a). Each ESM also shows consistency in the
relative declines under the four future climates (Figure S1a). The decline of wind driven transport
accounts for 47% (38% - 65% range) of the decline of Wind+Upwelling ITF transport under G6sulfur
during 2080-2100, and its ensemble mean wind driven transport volume is significantly lower than that
under SSP5-8.5 (Table 2). The ensemble mean ITF transport by buoyancy forcing all have significant
declining trend under the future climate scenarios but the differences are not generally significant (Figure
2d, Table 2), which is different from the transport change calculated using the wind driven and upwelling
contributions.

**Table 2**
The differences in monthly ITF Transport (2020-2100)[a] and its components according to the different
methods; Wind is the ITF transport derived from Island Rule and used in the Amended Island Rule;
Upwelling is the area integral of Pacific upwelling rate at 1500 m used in the Amended Island Rule;
Wind+Upwelling is the ITF transport calculated by Amended Island Rule; Buoyancy is the ITF transport
by buoyancy forcing and used independently of the other two components. Unit: Sv ($1Sv = 10^6 \, m^3/s$)

| Differences | Wind | Upwelling | Wind+Upwelling | Buoyancy |
|---|---|---|---|---|
| G6solar – SSP2-4.5 | 0.02 | **0.33** | **0.35** | -0.06 |
| G6sulfur – SSP2-4.5 | **-0.96** | **0.53** | **-0.44** | -0.21 |
| G6solar – SSP5-8.5 | **0.23** | **0.4** | **0.63** | -0.15 |
| G6sulfur – SSP5-8.5 | **-0.75** | **0.59** | **-0.16** | **-0.3** |
| G6sulfur –G6solar | **-0.98** | **0.19** | **-0.79** | -0.15 |


[a]The end dates of the G6solar and G6sulfur of MPI-ESM1-2-HR are 2099 and 2089, respectively, and
those of MPI-ESM1-2-LR are both in 2099. Values in bold are significant at the 95% level according
to the Wilcoxon signed-rank test.


The decline in ITF transport via upwelling in future relative to present under all scenarios is illustrated
in Figure 3. During the historical period, the zonally integrated, starting at of 44°S and proceeding
northward until 60°N, upwelling contributions to ITF transport in the Pacific Ocean steadily accumulate
when progressing from southern latitudes until about 20°N. Latitudes further north make little
contribution and accumulated upwelling is then fairly constant. This pattern changes in all future climate
scenario simulations. The Pacific upwelling contributions to transport volume accumulate steadily, but
slower with latitude than under the historical simulation, until to just north of the equator (2°N), and then,
after a small decrease rapidly accumulates over a few degrees of latitude. North of 20°N, the integrated
upwelling declines. Differences in ocean upwelling velocity under different scenarios are not significant
in the Pacific, except in the western boundary current region. Starting from 20°N, the wind stress in the
western boundary current region decreases, the upwelling of seawater weakens, (Figure 5), resulting in
a reduced upwelling contribution in the future scenario. Between 44°S and 15°S, the zonal cumulative
transport curves under SSP2-4.5 and G6solar are relatively similar. The integrated upwelling under the
G6sulfur scenario transitions from the smallest of the four future scenarios between 44°S and 20°S to the
largest a few degrees north of the equator (Figure 3).

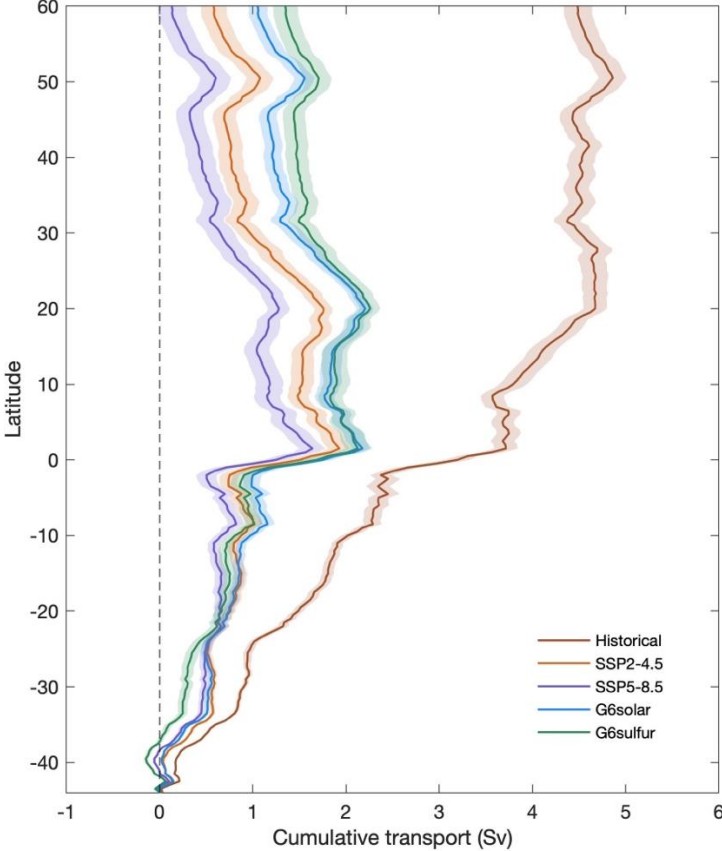


**Figure 3**. Multi-model ensemble mean zonal cumulative transport by Pacific upwelling north of 44°S

during the historical simulation (1980-2014) and under the four future scenarios (2080-2100), shadings

show the standard error.


**4.2 ITF by geoengineering type**

**4.2.1Wind stress**

Godfrey et al. (1993)suggested that the Indonesian throughflow originates in the South Pacific, where

the South Equatorial Current retroflects into the North Equatorial Countercurrent and enters the

Indonesian Sea via the Mindanao Current. Wind stress curl is determined by the components of the wind

stress vector and drives the ocean circulation (Gill and Adrian, 1982). Figure 4a shows the mean wind

stress and wind stress curl in the historical period (1980-2014), and the wind stress curl is positive at low

latitudes in the South Pacific, causing mass transport to the north. In the South Pacific under the SSP2-

4.5 scenario during 2080-2100, the wind stress curl in the middle latitudes is stronger than in the historical

period, while that at low latitudes and along the west coast of South America it is weaker than in the

historical period (Figure 4a). The SSP5-8.5 scenario anomalies relative to the historical period are similar
but extend over a larger region and have larger amplitude (Figure 4b). Net ITF transport volume under
SSP5-8.5 is lower than the historical, which is consistent with the difference in wind stress curl between
the simulations. There is no significant difference in wind stress curl between G6solar and SSP2-4.5 in
mid latitudes, and the difference in low latitudes is relatively small (Figure 4c). The wind stress curl
under G6solar is slightly weaker at mid latitudes and slightly stronger at low latitude than with SSP5-8.5
(Figure 4d). Differences between wind stress curl under G6sulfur and SSP2-4.5 scenarios are mainly in
the mid latitudes, near the equator and the west coast of South America (Figure 4e), which are related to
the wind driven ITF transport changes. In contrast, the significant differences between the wind stress
curl under G6sulfur and SSP5-8.5 are mainly in the northeast of the South Pacific, and the wind stress
curl under G6sulfur is stronger than that under SSP5-8.5 (Figure 4f). The wind stress curl at the inlet of
the ITF is significantly weakened under the G6sulfur scenario compared with the two SSPs scenarios.




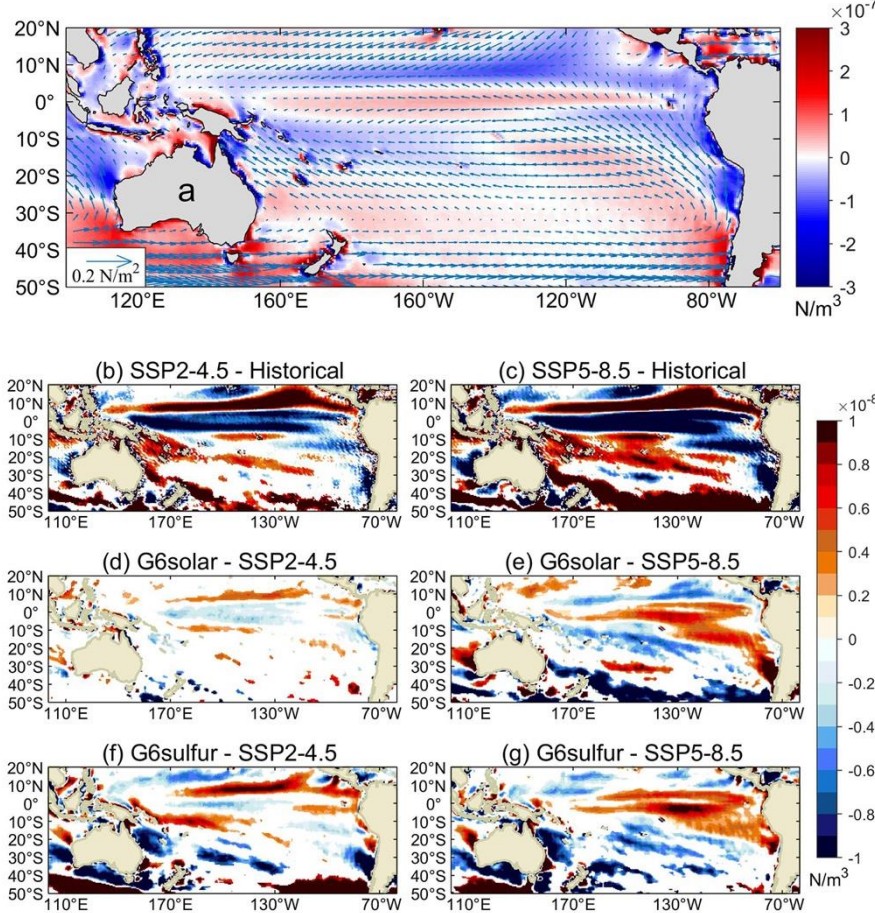

**Figure 4.** The multi-model mean differences in wind stress curl (a)the historical mean and the arrows show the wind stress, (b) SSP2-4.5 and historical, (c) SSP5-8.5 and historical, (d) G6solar and SSP2-4.5, (e) G6solar and SSP5-8.5, (f) G6sulfur and SSP2-4.5, (g) G6sulfur and SSP5-8.5. The historical period is 1980-2014, and the future scenarios period is 2080-2100. Regions where differences are not significant at the 95% level by the Wilcoxon signed-rank test are masked in white. Fig. S3 shows the ITF inlet region around the Indonesian archipelago in more detail.

The multi-model average ITF transport between G6 scenarios and SSPs scenarios shows significant differences during 2020-2100 (Table 2). Differences in wind-induced ITF transport from SSP2-4.5 are smallest with G6solar (Table 2) and are not significantly different in every ESM (Table S1). Differences between SSP5-8.5 and G6solar are the same sign for wind and upwelling forcings, contributing to larger differences in the amended island rule Wind+Upwelling transport. With G6sulfur, differences in wind and upwelling forcing differences from SSP5-8.5 are oppositely signed, and the net transport difference is quite small, but still significant for the six models ensemble (Table 2). Differences in the ITF defined

by buoyancy are only significant for G6sulfur-SSP5-8.5.


**4.2.2 Upwelling**
The spatial pattern of upwelling velocity at 1500 m in the Pacific under present day conditions is for
strong upwelling at the equator, weak upwelling in the interior, and mixed up- and down-welling along
the ocean boundaries (Feng et al., 2017). In future greenhouse gas climate scenarios, the main factor
affecting ITF transport is net upwelling in the Pacific Ocean (Feng et al., 2017; Sen Gupta et al., 2016).
Spatial patterns of upwelling changes are shown in Figure 5.

Much of the ocean shows no significant changes in upwelling velocity, but the western boundaries differ
significantly from the historical in both SSP scenarios (Figure 5a,b), and under SSP5-8.5 there is also a
significant upwelling in the equatorial eastern Pacific. The western boundary currents are an important
source of ITF gradient differences in wind stress that drive ocean currents (Hu et al., 2015), and these
gradients remain present at great depth in the western boundary current region.

The difference of upwelling velocity between G6solar and SSP2-4.5 scenarios is insignificant almost
everywhere (Figure 5c) once again illustrating the similarities between the solar dimming experiment
and its target SSP2-4.5 scenario. Differences from SSP5-8.5 are significant mainly along the extratropical
western ocean boundaries. The SAI experiment is clearly different from the solar dimming outcome.
G6sulfur differences from the SSP scenarios are clearly larger than those for G6solar and are greater in
the extra-tropics than in the tropics. The pattern of changes in upwelling anomalies for G6sulfur-SSP2-
4.5 is similar but of opposite sign to G6solar-SSP5-8.5 (Figure 5e), while differences for G6sulfur and
SSP5-85 are similar or slightly smaller than differences from SSP2-4.5 (Figure 5f).

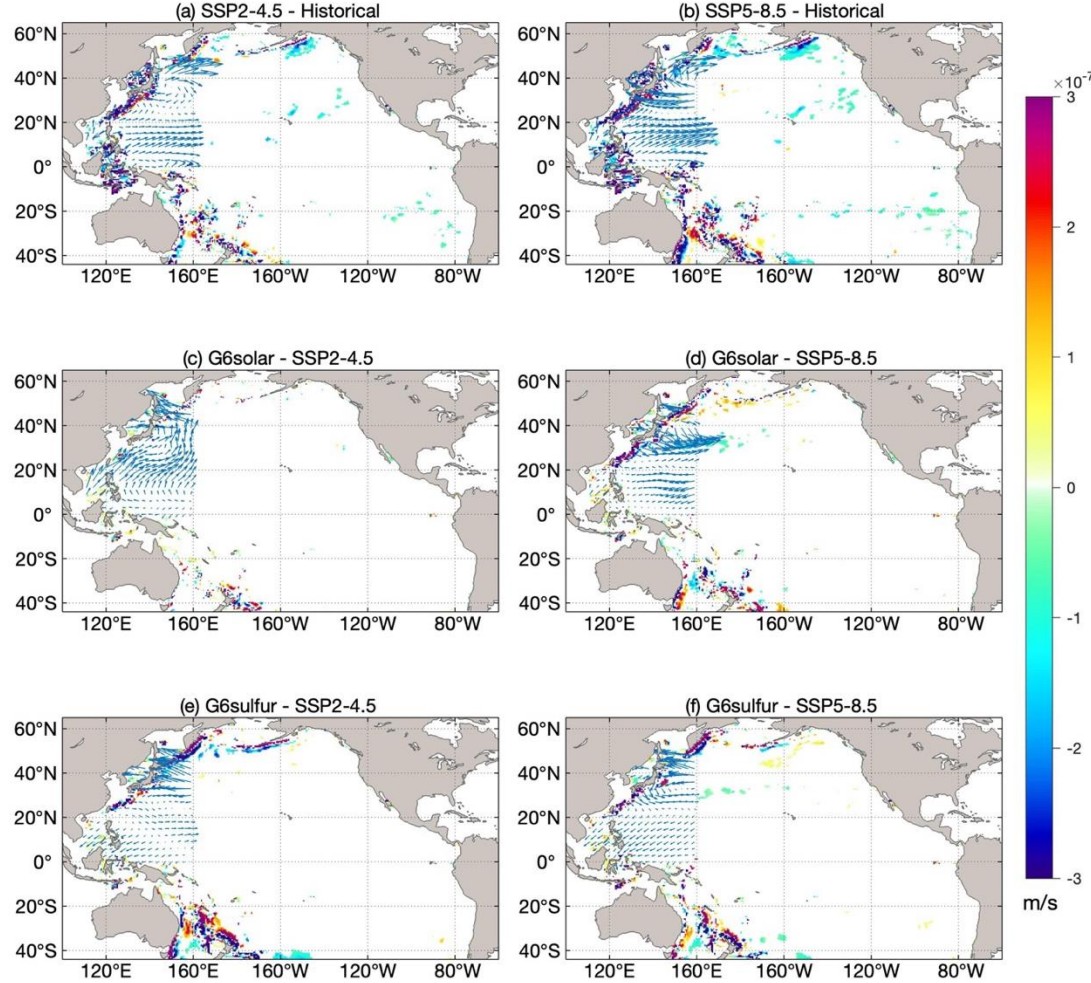

**Figure 5.** Changes in the multi-model ensemble mean upwelling velocity at 1500m (blue indicates increased upwelling, red indicates relative downwelling) and wind stress difference (arrow) for (a) SSP2-4.5 and historical, (b) SSP5-8.5 and historical, (c) G6solar and SSP2-4.5, (d) G6solar and SSP5-8.5, (e) G6sulfur and SSP2-4.5, (f)G6sulfur and SSP5-8.5. The historical period is 1980-2014, and the future scenarios period is 2080-2100. Regions where differences are not significant at the 95% level by the Wilcoxon signed rank test are masked in white.

### 4.2.3 Seasonality

Seasonal patterns in ITF are important and reflect changes in position of the two main precipitation convergence zones across the region. Model simulations show that decreases in ITF transport in April-May and October-November, and their recovery are due to the upper ocean changes associated with the Rossby waves in the Pacific Ocean, and that the seasonal ITF transport is closely related to wind

variations in the Pacific and Indian Oceans (Shinoda et al., 2012). The seasonal wind-driven ITF transport
is maximum in JJA and minimum in DJF under different scenarios (Figure 6), which is consistent with
the result by Wyrtki (1987). However, the differences between the G6 scenarios are largest in DJF and
MAM, and these seasons are also when all 4 future scenarios are most different from the historical
simulation.

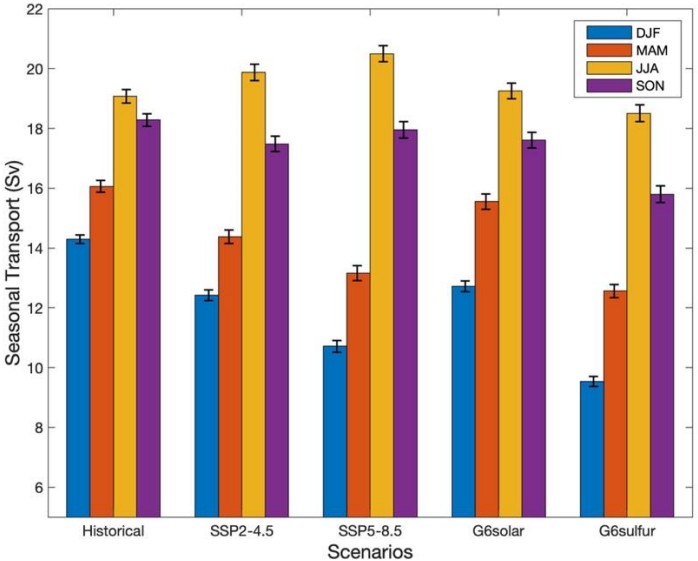


**Figure 6.** The ensemble mean seasonal wind-driven ITF transport and the standard error under the
historical period (1980-2014) and future scenarios (2080-2100).


The South Pacific Convergence Zone (SPCZ) is a strong rainfall and convection zone extending from
the equator to the subtropical South Pacific, which is generated by the low-level convergence between
the northeast trade wind and weaker westerly wind (Vincent, 1994). The SPCZ is clearest in December-
February (DJF), the Southern hemisphere summer, and is marked in the top row of Figure 7. The annual
wind stress curl differences between G6solar and SSP2-4.5 are small, but the seasonal variation
difference in some regions is significant. Under G6solar, compared with SSP2-4.5, the wind stress curl
near the equator is weakened in DJF. In March to May (MAM), the wind stress curl in the middle and
low latitudes of the southern hemisphere is generally enhanced. SSP5-8.5 has significantly lower wind
stress curl in the SPCZ region relative to G6solar in DJF. In MAM, their differences are mainly in the
mid latitudes. From June through November (JJA and SON), wind stress curl under SS5-8.5 is significant
lowered between 30 °S and 50 ° S. In contrast G6sulfur shows significant increase in the SPCZ region
in DJF, and a significant decrease the south of SPCZ region in JJA relative to SSP2-4.5. There are large
differences in the ocean northeast of New Zealand with the sign reversing from MAM to JJA. Differences
between G6sulfur and SSP5-8.5 are not very much bigger than from SSP2-4.5, and the patterns are quite
similar. The wind stress curl in the SPCZ region and its extension southeastwards is significantly
weakened under G6sulfur relative to both SSP scenarios in DJF. In JJA the region with decrease in wind
stress curl east from New Zealand is slightly larger relative to SSP5-8.5 and SSP2-4.5.

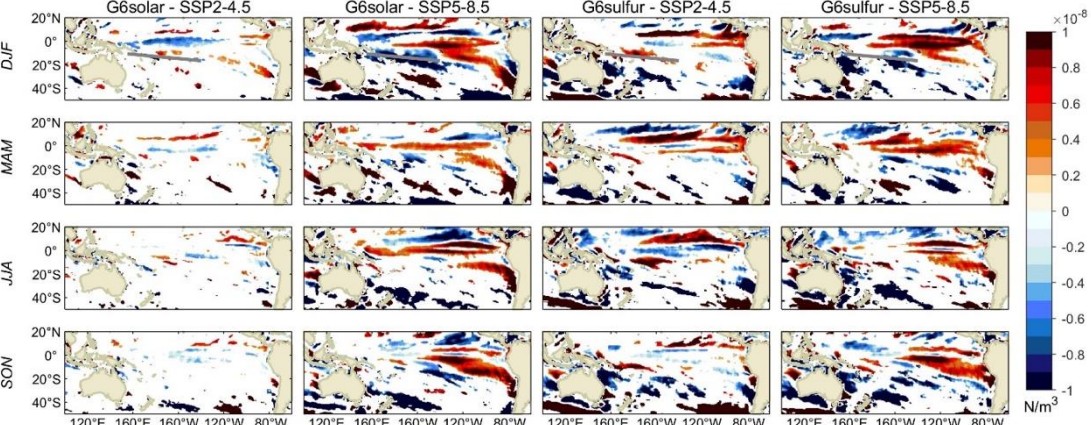


**Figure 7.** Seasonal ESM ensemble mean spatial differences (G6solar – SSP2-423 4.5, G6solar – SSP5-
8.5, G6sulfur - SSP2-4.5, G6sulfur – SSP5-8.5) of the wind stress curl during 2080-2100. The white
lines in each panel of the top row marks the mean the position of the South Pacific Convergence Zone
(SPCZ) in DJF based on the CMIP6 multi-model mean (Brown et al., 2020). Regions where
differences are not significant at the 95% level by the Wilcoxon signed rank test are masked in white,
significant differences are larger than $|0.5\times10^{-8}|$ Nm$^{-3}$

**4.3 ITF and ENSO**
The wind driven ITF transport estimated using the six CMIP6 models historical scenario is well within
the range of 11-20 Sv, found from 22 CMIP5 models (Sen Gupta et al., 2016). These model estimates
tend to slightly overestimate ITF compared with observed ITF (15±3 Sv) since Godfrey's Island Rule
ignores friction due to real ocean topography  (Feng et al., 2005; Wajsowicz, 1993). The rather large
interannual and decadal variations in the ITF (amounting to several Sv) are mainly influenced by the
Pacific and Indian Ocean winds. There is an observed relationship between ITF transport and the El
Niño-Southern Oscillation (ENSO), with stronger transport during La Niña and weaker transport during
El Niña, with ITF variability lagging ENSO variability by 8-9 months (England and Huang, 2005;
Meyers, 1996).

We seek relationships between ITF and ENSO variability using a wavelet coherence analysis (Grinsted
et al., 2004) of Nino3.4 and the wind-driven ITF anomaly. This method examines correlations and phase
between two time series, and is useful in exploring potential causality relationships (e.g. Grinsted et al.,
2004; Xia et al., 2023). Since the models are not adjusted to match observations, the natural variability
in the oceans is not synchronized, and so a multi-model ensemble will not show useful phase relationships,
so instead we show just the CESM2-WACCM model in Fig. 8, and the other models in Fig. S4. Figs 8
and S4 show obvious annual coherence for all models as could be expected as both time series have clear
seasonality, but this is not actually significant against the randomized phase Fourier background
hypothesis. There are multi-year significant power episodes in all models, though there are no significant
differences in power between the scenarios at any band between annual and decadal.  The two appear in
anti-phase (Figure 8) in line with observed stronger transport during La Niña and weaker transport during
El Niña. At the same time, ITF variability also lags behind ENSO on the whole, but there are differences
among different models.

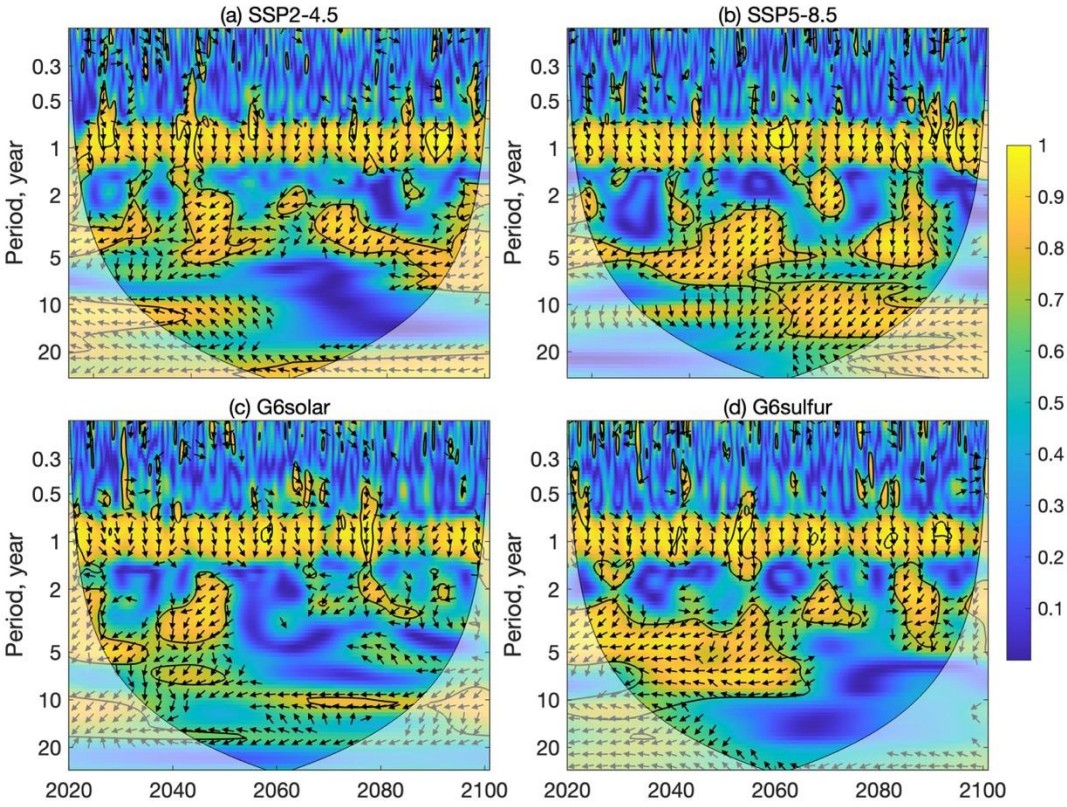

**Figure 8.** The squared wavelet coherence between the Nino3.4 (representing ENSO) and the wind-driven ITF transport monthly anomalies under the two SSPs (2015-2100) and two G6 (2020-2100) scenarios in CESM2-WACCM model. The 95% significance level above the background of 1000 Monte-Carlo ensemble of series of identical mean and standard deviation with identical power spectra but phase-randomized Fourier noise (chosen instead of the usual first order autoregressive null hypothesis here because of the strong annual signal; Xia et al. (2023)), is represented by a thick contour line. The arrows indicate the relative phase relationship, that is, in-phase points to the right, anti-phase points to the left, the arrow up indicates that the ITF anomaly leads ENSO by 90°, and a down arrow indicates that the ITF anomaly lags ENSO by 90°. The other models are shown in Fig. S4.

**5. Summary and Discussion**

The six ESM we use concur on weakening of ITF transport in all future scenarios. That is SRM cannot restore the ITF to its historic levels (Table 2, Fig 2). This contrasts somewhat to the changes simulated in the AMOC under SRM with GHG forcing, where it seems that SRM can partly reverse the slow down in AMOC induced by GHG forcing, reducing impacts from around 35% to 24% (Muri et al., 2018; Tilmes et al., 2020; Xie et al., 2022). This illustrates the important regional variability in responses to SRM.

Weakening of the ITF transport appears in all future scenarios, both with pure GHG forcing, and

combining GHG and SRM strategies. The ITF transport changes are defined almost totally (around 90%)
by significant differences in Pacific upwelling (Figure 2a and 2b). This is consistent with the conclusion
that the weakening trend of ITF under global warming predicted by high-precision ocean models is not
directly related to the change of Pacific trade winds but to the reduction of Pacific deep-sea upwelling
(Feng et al., 2017). On centennial scales, the decrease of the net deep ocean upwelling in the tropics and
the South Pacific, especially the changes in the western boundary current system is what determines ITF
transport. Buoyancy forcing can only estimate the interannual variation of the ITF, and our study
supports the utility of the Amended Island Rule in estimating centennial changes in ITF transport. The
Island rule was specifically formulated considering the difficulties in measuring flow in complex
topography. Instead, the Sverdrup theory of wind forcing was utilized, allowing much larger scale
observations to provide useful estimates of ITF. This methodology should also be suitable for the global
models we have analysed here. This contrasts with the relatively small regions of the DBP (Fig. 1), that
may not be consistently captured in the global models we analyze.

Sen Gupta et al. (2021) note that projected weakening of the ITF and differences between ESM can be
explained by changes in large-scale surface winds. This contrasts with our findings where changes in
wind driven transport are not significantly different between models, but instead upwelling in the
extratropical western boundary zones dominates changes between scenarios. However, western boundary
currents are deep and narrow and differ from the shallow and wide eastern boundary currents. The tropics
experience weaker (and reversed) trade winds from those that dominate the extratropical regions. The
geographical differences in upwelling suggest that wind changes are driving the overall changes in ITF
via upwelling regions, and so in effect supporting the conclusion of Sen Gupta et al. (2021) that
differences in future surface winds explain most of the differences in future large scale current systems.

SSP2-4.5 global radiative forcing was the design target of the G6 experiments despite GHG
concentrations being at SSP5-8.5 levels. The difference in wind stress curl between G6solar and SSP2-
4.5 indicates that the SD experiment performs better at reversing GHG induced changes in Pacific wind
than G6sulfur. The G6sulfur SAI experiment leads to a significant change in the winds in mid and low
latitude Pacific Ocean, which results in even lower estimated ITF transport than under the high GHG
SSP5-8.5 forcing alone. Furthermore, G6sulfur also impacts deep ocean upwelling especially in the
extratropical western boundary current region, such that the ITF transport during the 21$^{st}$ century under
the G6sulfur scenario is slower than that under the G6solar scenario. The G6 scenarios do not affect low
latitude western boundary currents and upwelling, for example the upwelling near the Mindanao current
is unaffected while the upwelling along the Kuroshio current is apparently displaced in both G6
experiments. The ITF transport under the SD experiment was stronger than under the SAI experiment
and even higher than its target SSP2-4.5 scenario level at the end of the 21st century.

Changes in circulation in the future will have important impacts on aquatic ecology and fisheries (Dubois
et al., 2016). In fact, the population in Indonesia's coastal areas, especially those in the islands through
which the ITF passes, are highly dependent on fisheries and hence, the changes in ITF under both pure
GHG and mixed GHG and SRM scenarios will have important local implications on the livelihood and
ways of life of the local populations. Seasonal variations in ITF transport reflect important processes in
the tropical convergence zones, and these are clearly impacted by all 4 future scenarios in generally subtle
ways. But the largest differences are seen between the two most challenging scenarios to simulate –
SSP5-8.5 and G6sulfur. Despite the large size of perturbation that these forcings apply in the simulations,
and the differences between climate models in parameterizing the SAI schemes, the finding are rather
robust in the changes of winds in all seasons in the Pacific Ocean and Maritime Continent.

SAI is a far more feasible method of SRM than SD (Shepherd, 2009), but it produces far larger
differences in various climate fields from GHG and historic simulations than does SD (Visioni et al.,
2021), and far larger across-ESM differences as the models process the aerosol impacts in varied ways
(Visioni et al., 2021). The differences in winds noted in G6sulfur likely arise from differences in
stratospheric heating due to the sulfur aerosols that then drive tropospheric circulation changes (Visioni
et al., 2020).

Although ESM can provide reliable predictions of the ITF transport, the accuracy of global meso- and
small-scale spatial and seasonal changes remains an issue. These relatively small-scale differences are
potentially more important for local impacts than differences in larger scale or annual changes. These
aspects will need to be explored using impact models tailored to the region, ideally through initiatives
focused on the Global South like the Degrees Initiative (https://www.degrees.ngo/) and addressing
concerns raised by local rightsholders.

**Code and data availability**
All model data used in this work are available from the Earth System Grid Federation (WCRP, 2022;
https://esgf-node.llnl.gov/projects/cmip6, last access: 3 July 2022).
**Author contributions**
JCM conceived and designed the analysis. CS collected the data and performed the analysis. CS and
JCM wrote the paper. All authors contributed to the discussion.
**Competing interests**
The contact author has declared that neither they nor their co-authors have any competing interests.
**Financial support**
This research has been supported by the National Key Research and Development Program of China
(grant nos. 2021YFB3900105), State Key Laboratory of Earth Surface Processes and Resource Ecology
(2022-ZD-05)and Finnish Academy COLD Consortium (grant no. 322430).

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
