# Peer review of "The Indonesian Throughflow Circulation Under Solar"

_Earth System Dynamics, 2023_

## Author Comment (AC1)

In this document text in blue is from the referee, text in black is our response and text in red are modifications for the manuscript.

Review of "The Indonesian Throughflow Circulation Under Solar Geoengineering" by Chen et al

In this manuscript, the authors analyse the changes in the ITF in two sets of GEOMIP6 simulations, with solar dimming and with stratospheric aerosol injection. They find that the major changes are due to upwelling in the Pacific Ocean, and not so much due to changes in wind stress.

This is in principle an interesting result, but I wonder whether it warrants a full manuscript by itself. The results are relatively 'thin', and there is very limited connection to how the changes in ITF impact other parts of the tropical ocean systems. I therefore encourage the Editor to carefully consider whether there is enough 'scientific meat' to the research question of ITF changes under geoengineering scenarios.

If the Editor does find the Research Question sufficiently relevant, then I have a few major concerns that I think should be addressed before the manuscript is ready for publication

1. The authors state that the ITF flow is too complex to be measured directly, but that is only (to some extent) true in observations. In model simulations, it is fairly trivial to simply integrate zonal and meridional transports, even more so in the coarser-resolution simulations of GEOMIP than in high-res simulations. I therefore really don't understand why the authors need to invoke buoyancy- and wind-stress based proxies of transport, if they could also measure transport directly

Reply: At present, only INSTANT has conducted comprehensive measurements of the ITF transport between 2004 and 2006. Observations are apparently not easy or trivial. In addition, our study focuses on the effects of geoengineering on ITF transport under global warming, which cannot be observed. The models even at 1 degree resolution are too coarse to capture the many small-scale features of the channels and the complex bathymetry. Obviously if it were possible to accurately compute ITF by the referee suggests, then there would be no need of the other methods that are actually used. So, we use accepted methods (wind and buoyancy, etc) to estimate the ITF transport under the future climates.

2. While relations with ENSO and other climate variability modes is discussed in the last section, there is no analysis of it. I'm surprised, as the effect of Geoengineering on ENSO is one of the many outstanding concerns. The argument that ENSO-analysis can't be done because the models are unforced is not very strong; there have been plenty of CMIP ENSO analyses.

Reply: Agreed, and so we added some relevant analysis. There are no statistically significant relationships between the models proxy of ENSO (Nino3.4) and the changes in wind stress. This null result is not very surprising considering that ENSO is driven by more than wind forcing, and indeed the interplay between winds and ocean temperatures are important and various timescales.

[Figure]

Figure; Scattergram of the wind driven ITF transport and Nino3.4 index under the future scenarios (2080-2100).

So we performed a wavelet coherence analysis as well.

[Figure]

**Figure 8.** The squared wavelet coherence between the Nino3.4 (representing ENSO) and the wind-driven ITF transport monthly anomalies under the two SSPs (2015-2100) and two G6 (2020-2100) scenarios in six models. The 95% significance level above the background of 1000 Monte-Carlo ensemble of series of identical mean and standard deviation with identical power spectra but phase-randomized Fourier noise (chosen instead of the usual first order autoregressive null hypothesis here because of the strong annual signal; Xia et al. (2023)), is represented by a thick contour line. The arrows indicate the relative phase relationship, that is, in-phase points to the right, anti-phase points to the left, the arrow up indicates that the ITF anomaly leads ENSO by 90°, and a down arrow indicates that the ITF anomaly lags ENSO by 90°.

: From the wavelet coherence analysis (Grinsted et al., 2004) of Nino3.4 and the wind-driven ITF anomaly, the obvious annual power is easily seen, but is not actually significant against the randomized phase Fourier background hypothesis. There are multi-year significant power in all models, though there are no significant differences in power between the scenarios at any band between annual and decadal. The two appear in anti-phase (Figure 8) in line with observed stronger transport during La Niña and weaker transport during El Niña. At the same time, ITF variability also lags behind ENSO on the whole, but there are differences among different models.

.

References

Grinsted, A.  J. C. Moore, S. Jevrejeva 2004 Application of the cross wavelet transform and wavelet coherence to geophysical time series, Nonlinear Processes in Geophysics, 11, 561-566

Xia, Y D.E. Gwyther, B. Galton-Fenzi, E.A. Cougnon, A.D. Fraser, J.C. Moore, 2023 Eddy and tidal driven basal melting of the Totten and Moscow University Ice Shelves, Frontiers in Marine Science, 10 https://doi.org/10.3389/fmars.2023.1159353

3. There is very little mention of the fidelity/skill of the GeoMIP simulations in this region. Only the mean transport is compared, but what about other EOVs like SST etc?

Reply:Yes we have not discussed this in any detail because we always look at anomalies and relative changes in ITF, winds, upwelling etc. There are certainly differences between models and observations that could be bias corrected e.g. as in Kuswanto, et al. (2021), but here we discuss relative differences in future scenarios rather than between compound indices such as apparent temperature.

Reference

Kuswanto, H., B. Kravitz, B. Miftahurrohmah, F. Fauzi, A. Sopahaluwaken, **J.C. Moore** 2021 Impact of solar geoengineering on temperatures over the Indonesian Maritime Continent *International Journal of Climatology* JOC-20-0686.R3 https://doi.org/10.1002/joc.7391

4. On line 238, the authors compare the wind-driven ITF of Fig 2a to the INSTANT observations of Sprintall et al; but these direct observations also include the buoyancy component so should be compared to Fig 2c; in which case the agreement is much poorer.

Reply: The comparison is between ITF estimated by the Island rule forcing and observations. The buoyancy model is entirely a separate method of estimating ITF, and it can been seen that the ITF estimated using buoyancy is far lower than using the Island rule (cf. 2b or 2c with 2d). The buoyancy method (2d) is not designed to be used together with wind (2a, the Island rule) or wind+upwelling (2c, Amended Island rule), and so the panels in figs 2a,c and 2d represent different models of how the observations can be simulated.

Added text: The multi-mean ITF transport simulated by buoyancy forcing is 7.3 Sv in the historical period, which is less than that by wind driven and only half the transport observed during INSTANT (Sprintall et al., 2009), and there is large across-model variability (Figure S2).

1. The explanation of the results in terms of climate physics is relatively limited. Most of the arguments in e.g. lines 313-334 are fairly handwaving and/or descriptive and could be substantiated by more careful and quantitative analysis.

Reply: We added some quantitative analysis, such as trend values and significance of differences between scenarios. We also discuss the results in terms of seasonal changes in forcing such as wind stress curl

finding consistent relationships, and upwelling. These are the model fields used that can be tested. We find no significant relationships with ENSO, but that is not surprising considering the large differences across models in fidelity of ENSO variability.

-line 19L state that this is the ITF water transport

Reply: Done

- line 19: in which way 'similar'?

Reply: The wind-driven ITF transport under the G6sulfur scenario shows a significant downward trend, and starting from about 2050, the transport is even lower than that under the SSP5-8.5 scenario, where the 'similar' is the reduced transport.

: … But stratospheric sulfate aerosols affects winds more than simply "shading the sun" and hence reduces the water transport similar as we simulate for unabated greenhouse gas emissions..

- line 57: the word 'compensating' is very confusing here. Agulhas leakage and ITF don't compensate each other

Reply: rephrased to " The ITF helps supply the Agulhas current leakage from the Indian Ocean to the South Atlantic Ocean,"

- line 58: is 'flush' the best word here?

Reply: Yes, it's an abstract expression, but it doesn't affect understanding.

- line 65: the use of 'flux' and 'transport' in one sentence raises the question whether these concepts are the same or not

Reply: The ITF transport can also be said to be the mass or volume flux, and its units are the same.

- line 80: The flow in the ITF is grossly simplified here. I strongly encourage the authors to be a bit more specific about the different pathways; and/or to show a model domain?

Reply: Yes, here we mainly mark the pathways mentioned in this paper, but have added some other pathways.

**a) The wind stress integral path and buoyancy region**

[Figure]

**b) Topography of Indonesian Sea**

[Figure]

- line 84: which simulations are meant here?

Reply: Done, I have added the citation.

: Analyzing the water flux through the many shallow channels in the Indonesian archipelago is challenging, and many of these channels are not resolved in simulations with resolutions of a degree or so (Gordon et al., 1999) (Figure 1).

- line 99: how is this transport observed?

Reply: the INSTANT use the mooring placed at major inflow and outflow passages, and the ITF transport was estimated to be around 15 Sv during the 3-year period (2004-2006).

: INSTANT uses moorings deployed at the major inflow (Makassar Strait, Lifamatola Strait) and outflow passages (Lombok Strait, Ombai Strait and Timor Passage) of the ITF to estimate the ITF transport, resulting in a value of 15 Sv during 2004-2006.

- line 112: why are these particular methods unlikely to be ever done?

Reply: Reworded as:

These styles of SRM are known to produce over-cooled tropical oceans and under-cooled poles relative to global mean temperatures. However, other styles of injection strategies than the simple tropical site specified by G6 can produce simulated climates without these temperature biases (MacMartin and Kravitz, 2016)

- Lin 159: why are the G6 scenarios not particularly realistic?

Reply: Reworded as:

These styles of SRM are known to produce over-cooled tropical oceans and under-cooled poles relative to global mean temperatures. However, other styles of injection strategies than the simple tropical site specified by G6 can produce simulated climates without these temperature biases (MacMartin and Kravitz, 2016)

- line 185: is 'dormant' the right word?

Reply: Yes we think the 'dormant' is right, implies that at a certain depth the sea is no motion.

- line 224: It's unclear whether these variables are calculated as a function of time, or using time-mean fields

Reply: we use the monthly oceanic temperature and salinity to calculate the ITF transport as stated just before Table 1: We used monthly data from the first realization in each scenario

- line 227: is this lever of 1200 m also an appropriate choice here?

Reply: We follow the analysis of Andersson and Stigebrandt, (2005), we do not make a novel method.

- line 229: is this the difference in spatially averaged densities?

Reply: Yes, as we say "… the density difference between the DBP region ($9°$ S–$15°$ S, $100°$ E–$120°$ E) and the EIO region ($6°$ N–$6°$ S, $80°$ E–$100°$ E).

- line 268: what is meant with a 'scheme' here?

Reply: Rephrased as "where ITF is much closer that from the wind driven estimation method."

- line 448/449: refer to where this is shown in the analysis

Reply: In Fig 2 and table 2, it is clear that ITF slows significantly in all scenarios.

Modified text :The six ESM we use concur on weakening of ITF transport in all future scenarios. That is SRM cannot restore the ITF to its historic levels (Table 2, Fig 2). This contrasts somewhat to the changes simulated in the AMOC under SRM with GHG forcing, where it seems that SRM can partly reverse the slow down in AMOC induced by GHG forcing, reducing impacts from around 35% to 24% (Muri et al., 2018; Tilmes et al., 2020; Xie et al., 2022). This illustrates the important regional variability in responses to SRM.

---

## Author Comment (AC2)

Referee #1

In this document text in blue is from the referee, text in black is our response and text in red are modifications for the manuscript.

This study examines the changes to the Indonesian throughflow (ITF) under two solar radiation management (SRM) scenarios – Solar Dimming (SD) and stratospheric aerosol injection (SAI). The SRM outcomes are compared to the SSP2-4.5 and SSP5-8.5 scenarios. The study confirms previous work that shows the ITF decline in the 21st century is primarily driven by changes to Pacific Upwelling, with the tropical easterly wind changes a secondary component. Furthermore, SD shows a similar decline in the ITF to SSP2-4.5, but the ITF decline in SAI is still similar to SSP5-8.5 (despite the net anthropogenic radiative forcing in SAI being similar to SSP2-45). This is because in SAI, the wind-driven component also declines greatly.

The ITF is likely an important metric to track in SAI simulations. However, the climatical and biogeochem importance of ITF and how these change with respect to GHG forcing or SAI is not well established in this paper. Below, I suggest some broader ideas for the authors to address followed by line-by-line suggestions through Line 300 (for Line 300 to the end, please make similar adjustments as stated previously).

1. In the Introduction, the authors could better clarify the importance of the ITF, what drives the ITF, how these drivers are projected to change under GHG and thus why ITF is projected to decline, what does SD and SAI research show for Pac-Ind variability that is relevant to the current study, and clearly state the main objectives of the paper. Also, the Intro and Methods could better explain the different components of ITF – I don't know if Buoyancy is an additional component to wind and upwelling, or if buoyancy is another way to calculate the full ITF transport.

Reply: Thanks we have elucidated as specifically requested and clarified that buoyancy is a separate methodology

2. The significance of trends and changes to the ITF transport are not detailed in the first half of the Results. Furthermore, when Wilcoxon stats are introduced, it is unclear what data is being compared (the monthly model-mean from 202001-209912?) and how the authors are considering the multi-model spread which is quite large compared to the variability in the multi-model mean. Furthermore, for many ocean variables there is usually autocorrelation, please explain how this is addressed in the sample size.

Reply: The statistical methodology used is the Wilcoxon test which is non-parametric and compares time series by rank, hence autocorrelation is not relevant, and which has been used very widely in climate research. We have also added standard errors where useful assuming Normality, but all significance testing is done with the Wilcoxon between the paired samples in the 2 scenarios being compared. We have also taken care to specify which time periods are being compared.

3. For each Results section, the main points are lost. Rather than just listing features of the relevant figures, please note how these features impact the ITF and its components and how these changes are consistent with other studies or if they are unique to this study.

Reply: Hopefully we have done this satisfactorily after following the specific concerns raised

4. In total, the amount of typographical errors, incomplete or run-on sentences, paragraphs without a clear point, are unfortunately distractions to the reader. As a reviewer, it's hard to address the main points of a paper when the review process becomes just an edit of the text.

Reply: Sorry for the inconvenience and thank you for your patience. It is indeed annoying to deal with these kinds of errors.

As is, I cannot recommend this study for publication. Though, I hope my comments are useful and will aid in your next submission.

.

Line 15-20 – the short summary is not clear and lacks relevant details

Reply: I have modified the text.

The Indonesia Throughflow is an important pathway connecting the Pacific and Indian Oceans and is part of a wind-driven circulation that is expected to reduce under greenhouse gas forcing. Solar dimming and sulfate aerosol injection geoengineering may reverse this effect. But stratospheric sulfate aerosols affects winds more than simply "shading the sun" and hence reduces the water transport similar as we simulate for unabated greenhouse gas emissions.

Line 32 – include historical time period.

Reply: Done

Six model ensemble mean projections for 2080 - 2100 relative to historical (1980-2014) ITF are reductions of…

Line 36 – What's the error?  Bc if SSP2-4.5 is 23% down, and G6solar is 19% down and G6sulfur is 28% down, we are talking about a magnitude 4% and 5% from SSP2-4.5 – which may not be a sig difference.

Reply: The scenario differences in ITF transport are all significant at the 95% level according to the Wilcoxon signed-rank test as shown in table 2.

Six model ensemble mean projections for 2080 - 2100 relative to historical (1980-2014) ITF are reductions of 19% under the G6solar scenario and 28% under the G6sulfur scenario which compare with reductions of 23% and 27% under SSP2-4.5 and SSP5-8.5. Despite standard deviations amounting to 5-8% for each scenario, all scenarios are significantly different from each other ($p<0.05$) when taken over the whole 2020-2100 simulation period.

Line 38 – 38% - 65%  not "~"

Reply: Done

the G6sulfur experiment shows a large reduction in ocean surface wind stress forcing accounting for 47% (38% - 65% across model range)…

Line 45 – remove "of"

Reply: Done

The ITF brings about 15 Sv (1 Sv = 106 m3/s; ~10.7 to ~18.7 Sv during the INSTANT Field Program, 2004-2006) of warm and fresh water from the Pacific to the Indian Ocean.

Line 49-52 – What are some examples of the ITF role in global climate? Many who read this paper will not be experts of the ITF, or oceanography – I think the introduction needs to more clearly explain the climatical importance of the ITF, its projected changes under SSP, what that means for the climate and biogeochem, etc.

Reply: Added the details about the importance of the ITF for the climate, the prediction under ssp585 scenario and the potential impacts the climate.

The ITF also plays an important role in regulating global climate and biogeochemical cycles (Ayers et al., 2014; Hirst and Godfrey, 1994), for example the ITF may influence the El Nino-Southern Oscillation (ENSO) by altering the tropical–subtropical exchange, the structure of the mean tropical thermocline, and the mean sea surface temperature (SST) difference between the Pacific warm Pool and the cold tongue, (Lee et al., 2002), and in the supply of iron in the equatorial upwelling, maintaining biological production in the equatorial eastern Pacific (Gorgues et al., 2007). Sen Gupta et al. (2021) used 26 CMIP6 models to predict ITF weakening by 3 Sv (2.4-3.2 Sv model range) under the SSP5-8.5 scenario (the high greenhouse gas emission scenario) relative to 20th century historical means The decline in the ITF would lead to more heat to accumulate in the Pacific Ocean, which could alter tropical atmospheric-ocean interactions and contribute to extreme El Nino /La Nina events (Cai et al., 2015; Klinger and Garuba, 2016).

References:

Cai, W., Santoso, A., Wang, G., Yeh, S.-W., An, S.-I., Cobb, K. M., Collins, M., Guilyardi, E., Jin, F.-F., Kug, J.-S., Lengaigne, M., McPhaden, M. J., Takahashi, K., Timmermann, A., Vecchi, G., Watanabe, M., and Wu, L.: ENSO and greenhouse warming, Nat. Clim. Change, 5, 849-859, https://doi.org/10.1038/nclimate2743, 2015.

Klinger, B. A., and Garuba, O. A.: Ocean Heat Uptake and Interbasin Transport of the Passive and Redistributive Components of Surface Heating, J. Clim., 29, 7507-7527, https://doi.org/10.1175/JCLI-D-16-0138.1, 2016.

Line 84 – are you referring to GCM simulations with ~1 deg ocean?

Reply: Yes. …with resolutions of a degree or so…

Line 91 – what defines "high-resolution"

Reply: Here,1/10°. The model is the Ocean Forecasting Australia Model version 3 (OFAM3), a near-global 1/10° ocean general circulation model, we called it a "high-resolution model", consistent with Hayashida et al. (2020).

Reference:

Hayashida, H., Matear, R.J., Strutton, P.G. *et al.* Insights into projected changes in marine heatwaves from a high-resolution ocean circulation model. *Nat Commun* **11**, 4352 (2020). https://doi.org/10.1038/s41467-020-18241-x

This motivated Sen Gupta et al. (2016), and Feng et al. (2017) to propose the Amended Island Rule that modifies the Island Rule to include the estimated net Pacific upwelling contribution to ITF based on high-resolution (1/10°) ocean general circulation modelling

Line 106 – what is meant by "imperfectly"

Reply: By imperfectly we mean that the spatial and temporal pattern of radiative forcing with solar geoengineering is different from that produced by removing greenhouse gases. We add a citation (Kravitz et al., 2015), and note more details a little later "These styles of SRM are known to produce over-cooled tropical oceans and under-cooled poles relative to global mean temperatures,"

Lines 112-114 –What do you mean by "… these particular methods are unlikely to ever be done" --- I presume you mean this exact G6Sulfur SAI strategy is unlikely to be what is chosen if SRM is ever used in the future. This sentence can be re-worded to more clearly state that ongoing research is examining a multitude SAI strategies (how much, at what latitude, at which altitude, etc) to reduce the equator-to-pole cooling bias.

 Reply: Yes. Reworded as:

These styles of SRM are known to produce over-cooled tropical oceans and under-cooled poles relative to global mean temperatures. However, other styles of injection strategies than the simple tropical site specified by G6 can produce simulated climates without these temperature biases (MacMartin and Kravitz, 2016).

Line 114 – add "atmospheric" to read: Simulated tropical atmospheric circulation

Reply: Done

Simulated tropical atmospheric circulation…

Line 115 – replace ";" with "." And start new sentence: Under SD, the seasonal …

Reply: Done

systems are impacted under both GHG and solar geoengineering scenarios. Under SD, …

Line 115-123 – can you be more specific to the changes to tropical Pacific atmospheric circulation due to SD and SAI (or SRM in general)? And, it's not clear to me why citations about NA hurricane numbers and CAPE under SRM are relevant to this study.

Reply: We have added more details in the tropical circulation and explained why these are important in tropical cyclogenesis:

Both the Hadley and Walker circulations are different from the historical (Guo et al., 2018; Cheng et al., 2022). Impacts of SRM on the Walker circulation are modest compared with the Hadley cell but appear most obviously in relation to the South Pacific Convergence Zone (Guo et al., 2018), which is

relevant in the overall tropical Pacific atmosphere system that drives and interacts with the ITF. Greenhouse gas forcing is expected to cause an expansion of the Hadley circulation cells which may be asymmetric between northern and southern hemispheres (Staten et al., 2019). Both SD (Guo et al., 2018) and SAI (Cheng et al., 2022) reduce these greenhouse gas induced changes in the Hadley circulation, although again hemispheric differences remain, and in the Cheng et al., (2022) simulations, were associated with stratospheric heating and tropospheric temperature response due to enhanced stratospheric aerosol concentrations. The changes in stratospheric heating, the tropopause height, and tropical sea surface temperatures may be expected to impact tropical cyclogenesis, and this is consistent with reduction in North Atlantic hurricane numbers and intensity relative to GHG-only climates under SAI (Moore et al., 2015). However, there are differences between tropical basins in expected tropical cyclogenesis potential and significant differences in simulations between climate models (Wang et al., 2018). Potential energy available for extratropical storms is also consistently reduced under SRM relative to GHG forcing (Gertler et al., 2020). The reported impacts highlight the potential role of wind forcing in ITF.

Cheng, W. D. MacMartin, B. Kravitz, D. Visioni, E. Bednarz, Y. Xu, Y. Luo, L. Huang, Y. Hu, P. Staten, P. Hitchcock, J.C. Moore, A. Guo, X. Deng 2022 Changes in Hadley circulation and intertropical convergence zone under strategic stratospheric aerosol geoengineering. *npj Climate and Atmospheric Science* **5,** 32 https://doi.org/10.1038/s41612-022-00254-6

Staten, P. W., Grise, K. M., Davis, S. M., Karnauskas, K. & Davis, N. Regional Widening of Tropical Overturning: Forced Change, Natural Variability, and Recent Trends. *Journal of Geophysical Research: Atmospheres* **124**, 6104–6119 (2019)

Line 125-134 – I don't think this paragraph is necessary. Perhaps only a sentence is needed to state that the AMOC under SRM has been looked at. I would rather have another paragraph explaining tropical pacific/Indian ocean changes under SRM and more details about the ocean drivers of ITF. That said, Lines 446-452 compare the continual downturn of ITF to research that shows that AMOC can recover under SRM – this is a good comparison to make at this stage of the paper.

Reply: As the referee notes, we return to the AMOC in the discussion. As almost nothing has been done on oceanographic impacts of solar geoengineering, we felt it essential to introduce what has, and also consider a little on the differences between the types of SRM.

In general, the intro could use more details about atmo circulation and how that changes under GHG and SRM. More about ITF and what drivers can cause changes to it (bc many geoeng don't know ocean and ITF). For example, "this study will show how SRM can change these drivers of ITF". Some details are in the methods, but better placed in the introduction.

Agreed, and we hope the changes made above address this concern.

Line 136 - 137 – replace explore with examine and remove the clause "explore the drivers of these changes"

Reply: done

Line 137-138 suggestion: … 21st century and consider the transport differences between the GHG-forced scenarios (SSP2-4.5 and SSP5-8.5) and the SRM scenarios (G6solar and G6sulfur).

Reply: Actually, we have checked the drivers such as wind stress curl and the Pacific upwelling velocity. So I think we need keep the meaning, just remove the clause and modify the text.

In this study, we will examine the impact of SRM on the change of the ITF in the 21$^{st}$ century, and consider the transport and drivers differences between pure GHG climates representing moderate mitigation (SSP2-4.5) and no mitigation (SSP5-8.5); with solar dimming (G6solar) and stratospheric aerosol injection (G6sulfur) forms of SRM geoengineering.

Line 159 – why aren't G6 scenarios particularly realistic? And as a follow-up, why should the current study and its results be considered, if the simulations are not realistic?

Reply: we explain and expand as:

While the G6 scenarios are not particular realistic, for example they specify starting SAI in 2020 and specify a very simple tropical injection strategy, they do provide a usefully large SRM and GHG signal, and have been simulated by six CMIP6 generation models. This allows more robust findings of the general impacts of SAI, especially when considering aspects of the climate system that have not been addressed to date in geoengineering studies, such as the ITF.

Line 197-208 – This explanation of the GHG forced changes to deep ocean upwelling and their impact on ITF would be a good addition to the revised introduction.

Reply: We had similar text already in the introduction, and we do not want to have equations so early in the introduction. However, we rephrase and expand the key Introduction paragraph:

Analyzing the water flux through the many shallow channels in the Indonesian archipelago is challenging, and many of these channels are not resolved in simulations (Figure 1). This motivates the use of alternative methods of estimating ITF. Godfrey (1989) created the Island Rule to estimate flux based on Sverdrup theory (Sverdrup, 1947) analysis of Pacific wind stress. More recently, analysis of climate models revealed the importance of deep ocean circulation to the reduction of ITF transport under GHG forcing. The decline in ITF under GHG forcing could be due to both the weakening of trade winds in the Pacific, and deep ocean circulation changes (Feng et al., 2012; Hu et al., 2015). Interannual to decadal, as well as centennial dependence of the ITF on wind and upwelling was found with an eddy-resolving ocean model simulation (Feng et al., 2017). This led to Sen Gupta et al. (2016), and Feng et al. (2017) proposing the Amended Island Rule that modifies the Island Rule to include the estimated net Pacific upwelling contribution to ITF based on high-resolution (1/10°) ocean general circulation modelling.

An alternative mechanism for the ITF driver was proposed earlier by Andersson and Stigebrandt (2005). In this theory buoyancy forcing is more important than wind forcing in driving the ITF. The ITF variability is found from the baroclinic outflow of the Downstream Buoyant Pool (DBP) that extends over much of the North Australian Basin (Figure 1). Hu and Sprintall (2016) used this method with reanalysis products to produce ITF interannual variability in good agreement with the observed volume transports (2004–2006) from the INSTANT mooring array transport (Sprintall et al., 2009), although the average transport was smaller than the observed transport. While the evidence suggests that the Amended Island Rule explains ITF variability better than buoyancy, changes in buoyancy forcing may affect volume transport of the ITF on decadal scales under a changing climate.

Line 211-212 – replace "in most studies (",  with: … in previous studies (e.g., Clarke …

Reply: Done

Sea levels in the Pacific and Indian Oceans have been used to estimate the ITF transport in previous studies (e.g., Clarke and Liu, 1994; Potemra et al., 1997; Susanto and Song, 2015).

Line 213 – define steric sea level height

Reply: Done

Buoyancy accounts for high steric sea level (that is a volume increase due to lower density), in …

Line 214 – "should also drive", I'm not sure what is meant by this phrase

Reply: deleted phrase, so the full sentence is: Buoyancy accounts for high steric sea level (that is volume increase due to low density), in the North Pacific (Stigebrandt, 1984).

Line 217 – replace "sharp" with eastern?

Reply: replaced "sharp" with "abrupt eastern"

The sea level drop between Indian and Pacific Oceans occurs essentially at the abrupt eastern boundary of the DBP and is the source of buoyancy forcing.

Line 218-220 – here ITF transport is defined as the difference btw westward and eastward transport along the northern and southern flanks… just below ITF transport is defined by DBP and EIO density change (above by Sverdrup balance and Pacific upwelling).  Why all of these? What do each tell us about ITF?

Reply: we want to present an analysis of ITF using the 2 main theories driving it. Which are: 1) wind (+ upwelling), and 2) buoyancy. It appears that both may play roles on different timescales. We differentiate these two alternatives more explicitly now. In the Introduction:

An alternative mechanism for the ITF driver was proposed earlier by Andersson and Stigebrandt (2005). In this theory buoyancy forcing is more important than wind forcing in driving the ITF. The ITF  variability is found from the baroclinic outflow of the Downstream Buoyant Pool (DBP) that extends over much of the North Australian Basin (Figure 1). Hu and Sprintall (2016) used this method with reanalysis products to produce ITF interannual variability in good agreement with the observed volume transports (2004–2006) from the INSTANT mooring array transport (Sprintall et al., 2009), although the average transport was smaller than the observed transport. While the evidence suggests that the Amended Island Rule explains ITF variability better than buoyancy, changes in buoyancy forcing may affect volume transport of the ITF on decadal scales under a changing climate.

Line 236 – is wind driven ITF just the Island Rule transport calculation?

Reply: Yes, the wind driven ITF is calculated by Island Rule.

INCLUDE +- 1 SD with respect to ensemble members in all values. For the 2080-2100 minus historical period, we need to know significance! It's likely that there is lag-1 autocorrelation in these time series too and that needs to be accounted for when defining sample size. And trend values and significance.

Reply: Actually standard errors are more useful for determining significance, but the Wilcoxon test is our choice since it makes no assumption of Normality in the distributions. So we simplify the results as:

: Six model ensemble mean projections for 2080 - 2100 relative to historical (1980-2014) ITF are reductions of 19% under the G6solar scenario and 28% under the G6sulfur scenario which compare with reductions of 23% and 27% under SSP2-4.5 and SSP5-8.5. Despite standard deviations amounting to 5-8% for each scenario, all scenarios are significantly different from each other ($p<0.05$) when taken over the whole 2020-2100 simulation period.

: During the last 20 years of the 21st century, the simulated ITF transport using the Amended Island Rule is 27% $\pm$ 3% (standard error), under SSP5-8.5 (Figure 2c), with Pacific upwelling decline accounting for 76%$\pm$8% ($p<0.05$) of the total reduction. Both wind driven and upwelling contributions to ITF transport are slightly higher under SSP2-4.5 than under SSP5-8.5 during the same period, but the differences are small over the whole 2015-2100 period. The total ITF transport is reduced by 23%$\pm$2% (standard error, $p<0.05$) under SSP2-4.5 during the period of 2080-2100 relative to the historical period (13%~27% cross ESM range), …

: SAI and SD geoengineering methods have different effects on wind driven and upwelling contributions to ITF transport (Figure 2a, b). Under the G6solar and G6sulfur scenarios, the total ITF transport is reduced by 19%$\pm$1% and 28%$\pm$1% respectively during 2080 - 2100 relative to the historical period, of which the wind-driven ITF transport is reduced by 4%$\pm$1% and 16%,$\pm$1% and the upwelling transport volume is reduced by 76%$\pm$8% and 70%$\pm$10%, all differences are significant ($p<0.05$), Table 2.

Line 242-245 – What are the trend values (for all ITF transport components)? & Calculate the significance of the (linear) trend lines. Because it looks like there is a negative trend in the wind driven component for both SSP2-4.5 and SSP2-8.5, but it may not be significant.

Reply: We have added the trend values and the significance of the trend lines in figure 2.

The wind driven volume ITF transport has significant trends for all scenarios with smallest trends for the SSP scenarios (linear trends of lower magnitude than 0.02 Sv per year), while the upwelling contributions has obvious downward trends in all scenarios. These trends appear to be consistent, despite differences in estimated transport across models (Figure S1).

[Figure]

Figure 2. Six ESM ensemble mean ITF components under different scenarios, shadings show the standard deviation and the formula is the trend fitting results under different scenarios and the significant value (The ranges is 2015-2100 under two SSP scenarios and 2020-2100 under two G6 scenarios). (a) Sverdrup balance wind driven component. (b) Pacific upwelling north of 44°S. (c) Total ITF under the Amended Island Rule (eqn 2). (d) ITF transport by buoyancy forcing. Individual ESM results are shown in Figure S1.

[Figure]

Figure S1. (a) the time series of ITF transport in the six ESMs for wind driven component under different scenarios. (b) as Figure S1a for Pacific upwelling contribution. (c) as Figure S1a for total ITF transport under Amended Island Rule.

Line 250-252 – These values, as all values in this paper should have a statistic attached to them that details the model spread (easiest would be +- 1 SD wrt model spread); and if comparing between SRM and the SSPs there needs to be a significance test in order to attach some level of confidence to the findings.

Reply: Done, but we think standard errors and results from the non-parameteric testing in Table 2 are more useful than the standard deviation.

During the last 20 years of the 21st century, the simulated ITF transport using the Amended Island Rule is 27% ± 2% (standard error), under SSP5-8.5 (Figure 2c), with Pacific upwelling decline accounting for 76%±8% (p<0.05) of the total reduction. Both wind driven and upwelling contributions to ITF transport are slightly higher under SSP2-4.5 than under SSP5-8.5 during the same period, but the differences are small over the whole 2015-2100 period. The total ITF transport is reduced by 23%±1% (standard error, p<0.05) under SSP2-4.5 during the period of 2080-2100 relative to the historical period (13%~27% cross ESM range), …

Line 252-253 – how can both wind and upwelling contributions to ITF be higher? Their sum is ITF transport, so if one contribution goes up, the other must go down?

Reply: The ITF transport estimated by the Amended Island Rule is the sum of the wind and upwelling contributions, and the ITF changes over time.

Line 261-264 – Should the buoyancy forcing calculation of ITF be more similar to the results from the Amended Island Rule? It's not clear to me if this is a component of ITF or another way to measure ITF. I'm not sure what is the main point/conclusion of this paragraph

Reply: Buoyancy forcing is another way to measure ITF, we also want to examine how buoyancy forcing drive the ITF transport changes under future scenarios. Hu et al. used the buoyancy forcing scheme to get the ITF transport of 10 Sv, which is less than the observed transport and may be related to the dataset. In any case, the point of our results is the change in transport. This is an entry point.

Line 264 – "No obvious trend" – you need to calculate the trend and determine whether it is significant

Reply: Done.

The wind driven volume ITF transport has significant trends for all scenarios with smallest trends for the SSP scenarios (linear trends of lower magnitude than 0.02 Sv per year)…

Line 277-278 – It looks to me like the upwelling contribution to ITF transport is not much different between SAI and SD – again a statistical test will confirm this.

Reply: In fact there are significant differences (Table 2), we have modified the text.

SAI and SD geoengineering methods clearly have different impacts on wind driven contributions to ITF transport but smaller although still significant differences in upwelling (Figure 2a,b, Table 2).

Line 286-287 – it may be that G6sulfur total ITF transport averaged from 2080-2100 is lower than SSP2-8.5, but I doubt it is significant given the variability of the time series.

Reply: ITF transport averaged from 2015-2100 is significantly lower than SSP5-8.5, as shown in Table 2.

…, and its ensemble mean wind driven transport volume is significantly lower than that under SSP5-8.5 (Table 2).

Line 287-291 – again, the time series shows much variability in the multi-model mean (and certainly in the multi-model spread) that I don't think there is a significant difference between the buoyancy forced ITF transport in any of the scenarios.

Reply: We have done the significant test using the standard approach.

The ensemble mean ITF transport by buoyancy forcing all have significant declining trend under the future climate scenarios but the differences are not generally significant (Figure 2d, Table 2), which is different from the transport change calculated using the wind driven and upwelling contributions.

Figure 3 needs error bars. And it would be easier to quickly assess by using the same colors for each scenario as in Fig 2.

Reply: Done

[Figure]

**Figure3**. Multi-model ensemble mean zonal cumulative transport by Pacific upwelling north of 44°S during the historical simulation (1980-2014) and under the four future scenarios (2080-2100), shadings show the standard error.

Lines 293 – there needs to be a detailed explanation of what is causing the decline in upwelling relative to Historical – as well as why the upwelling becomes downwelling north of ~20N. In general, rather than state what the plot shows, state what the plot means in relation to the main points of the paper.

Reply: We supplement the wind stress difference map for the Northern Hemisphere western boundary current region in Figure5. The reduction of wind stress in the SSP2-4.5 and SSP5-8.5 scenarios compared to historical periods is likely to cause changes in upwelling and downwelling in the deep ocean. Of course, the changes in ocean water throughout the Pacific are complex, and here we mainly look at the western boundary area where there are significant differences in ocean velocity.

Differences in ocean upwelling velocity under different scenarios are not significant in the Pacific, except in the western boundary current region. Starting from 20°N, the wind stress in the western boundary current region decreases, the upwelling of seawater weakens, (Figure 5), resulting in a reduced upwelling contribution in the future scenario.

Figure 4 – insert a panel in the top left that is just the Historical mean wind stress curl so that it is more easy to interpret the changes from this mean state. You could also just make all non-significant

values white. Removing the stippling (and adjusting the saturated colorbar) will improve the figure's aesthetic.

Reply: Done

[Figure]

**Figure 4**. The multi-model mean differences in wind stress curl (a)the historical mean and the arrows show the wind stress, (b) SSP2-4.5 and historical, (c) SSP5-8.5 and historical, (d) G6solar and SSP2-4.5, (e) G6solar and SSP5-8.5, (f) G6sulfur and SSP2-4.5, (g) G6sulfur and SSP5-8.5. The historical period is 1980-2014, and the future scenarios period is 2080-2100. Regions where differences are not significant at the 95% level by the Wilcoxon signed rank test are masked in white.

Lines 327-330 – Can you explain how these changes in wind stress curl impact the ITF transport changes? – In general for this paragraph, it's unclear to me how these various changes to wind stress curl across the basin will impact ITF. I am unable to decipher the main point(s) of this paragraph.

Reply: The wind stress curl can affect the upwelling and downwelling of seawater, thus causing the mass transport of seawater (ITF). So here we compare the difference of wind stress curl.

Figure 4a shows the mean wind stress and wind stress curl in the historical period (1980-2014), and the wind stress curl is positive at low latitudes in the South Pacific, causing mass transport to the north.

Lines 346-354 & Table 2 – What is the acronym TRN? For the Wilcoxon signed-rank test, how do you account for model spread at each month? For example, I'm having difficultly being convinced that the 0.23 change from G6-solar to SSP5-8.5 is significant given that large multi-model variance

compared to the signal. Could you elaborate on this stat test and how the model spread is accounted for?

Reply: TRN is an acronym for transport, we remove it in table 2 as the differences are important in wind and upwelling and buoyancy. We use the monthly data to do the Wilcoxon signed-rank test. The test is a standard non-parametric test used for many climate time series.

Lines 368-384 – Please clarify the main point(s) of this paragraph and relate the significance of upwelling changes to ITF transport.

Reply: In the future climate, the main factor affecting ITF transport is net upwelling in the Pacific Ocean. According to our analysis, within the Pacific region, the differences under different future scenarios are concentrated in the western boundary current region.

Added text: In the future climate scenarios, the main factor affecting ITF transport is net upwelling in the Pacific Ocean.

Figure 5 – Just mask out in white the non-significant areas and increase the colorbar range so the stat sig areas are not saturated.

Reply: Done.

[Figure]

**Figure 5**. Changes in the multi-model ensemble mean upwelling velocity at 1500m (blue indicates increased upwelling, red indicates relative downwelling) and wind stress difference (arrow) for (a) SSP2-4.5 and historical, (b) SSP5-8.5 and historical, (c) G6solar and SSP2-4.5, (d) G6solar and SSP5-8.5, (e) G6sulfur and SSP2-4.5, (f) G6sulfur and SSP5-8.5. The historical period is 1980-2014, and the future scenarios period is 2080-2100. Regions where differences are not significant at the 95% level by the Wilcoxon signed rank test are masked in white.

Lines 394-420 & Figure 6 – Similar comments to above.

Reply: Done

[Figure]

**Figure 6.** The ensemble mean seasonal wind-driven ITF transport and the standard error under the historical period (1980-2014) and future scenarios (2080-2100).

: The seasonal wind-driven ITF transport is maximum in JJA and minimum in MAM under different scenarios (Figure 6), which is consistent with the result by Wyrtki (1987). However, the differences between the G6 scenarios are largest in DJF and MAM, and these seasons are also when all 4 future scenarios are most different from the historical simulation.

[Figure]

**Figure 7**. Seasonal ESM ensemble mean spatial differences (G6solar – SSP2-423 4.5, G6solar – SSP5-8.5, G6sulfur - SSP2-4.5, G6sulfur – SSP5-8.5) of the wind stress curl during 2080-2100. The white lines in each panel of the top row marks the mean the position of the South Pacific Convergence Zone (SPCZ) in DJF based on the CMIP6 multi-model mean (Brown et al., 2020). Regions where differences are not significant at the 95% level by the Wilcoxon signed rank test are masked in white, significant differences are larger than $|0.5 \times 10^{-8}|$ Nm$^{-3}$

Summary and Discussion – These paragraphs are easier to understand. To improve the clarity of the whole paper, these main points should be established as the goals of the paper in the Introduction and then addressed throughout the Results. As the paper stands now, it is difficult to determine the main points and how the results address these points.

Reply: thanks, we hope it is clearer now.

Line 467-469 – I missed the analysis where wind driven transport is compared between models? Given the multi-model range from Fig 2, it seems like the models would have some disagreement with respect to wind driven transport.

Reply: In Figure S1, we give the wind driven conveying capacity of each model. There are differences in values among different models, but the trends are similar.

---

## Author Response (AR2)

Rebuttal 19 October

The authors have done an excellent job with the revisions of the previous version of the manuscript, as well as the responses to review comments. I have a few additional comments, mostly related to clarifying things.

General comment: There are a few places where I think some clarification and caveating of the results would be appropriate. The simulations you use are for global-scale models, yet it is well known that in many cases finer-scale (often regional) models are necessary to capture transport through narrow straits. As such, your results are likely off (as is evidenced in lines 260ff), although the overall transport is not bad (lines 237ff). I think some more clarity would be helpful, in particular descriptions as to when we can likely trust the results. This is especially important because we don't have observations for geoengineering, so we need to know when the metrics are giving trustworthy answers.

There are several factors here that are playing a role. The Island rule was specifically formulated to take into account the difficulties in measuring flow in complex topography. Instead, as we explain the Sverdrup theory of wind forcing was developed, and this much larger scale methodology should also be suitable for the global models we have analysed here.

Some of the large differences between the observations and the method are probably because the buoyancy hypothesis for the ITF is incorrect, or possibly the relatively small regions of the DBP (Fig. 1) are not well capture in the global models. We point this out more explicitly in the Introduction: "In contrast with the reasonable agreement for the Amended Island Rule estimates of ITF, the alternative buoyancy method behaves much worse, indicating that the hypothetical forcing is not as good an explanation for ITF as the Amended Island Rule, or that the models used do not capture the specific details of the DBP. But although the Amended Island Rule matches the short duration of observed fluxes and variability better than buoyancy, it is possible that changes in buoyancy forcing may affect volume transport of the ITF on decadal scales under a changing climate and so we examine its changes under the geoengineering scenarios."

In the Summary we also add this text: "The Island rule was specifically formulated considering the difficulties in measuring flow in complex topography. Instead, the Sverdrup theory of wind forcing was utilized, allowing much larger scale observations to provide useful estimates of ITF. This methodology should also be suitable for the global models we have analysed here. This contrasts with the relatively small regions of the DBP (Fig. 1), that may not be consistently captured in the global models we analysed."

Figure 4: I like this figure, but it's hard to see what's going on. I'd like to see (in addition) a version of this figure that focuses on the ITF inlet. I'll let you decide which of the figures goes in the main article, supplement, etc.

We made a new plot and decided to have this in the supplemental material (Figure S3):

[Figure]

**Figure S3**. The ITF inlet region around the Indonesian archipelago in more detail than shown in Fig. 4. The multi-model mean differences in wind stress curl (a) the historical mean and the arrows show the wind stress, (b) SSP2-4.5 and historical, (c) SSP5-8.5 and historical, (d) G6solar and SSP2-4.5, (e) G6solar and SSP5-8.5, (f) G6sulfur and SSP2-4.5, (g) G6sulfur and SSP5-8.5. The historical period is 1980-2014, and the future scenarios period is 2080-2100. Regions where differences are not significant at the 95% level by the Wilcoxon signed-rank test are masked in white.

Section 4.2.2: I found this section to be written confusingly. The section jumps back and forth between topics and isn't clear about when you're looking at climate change vs geoengineering. I"d recommend some organization.

Agreed. We have broken the text into 3 paragraphs and generally clarified the structure.

Section 5: The first two paragraphs and Figure 8 aren't part of the summary. They're a new thing. I'd move these into their own subsection in Section 4.

Agreed, and the text is expanded for clarity.

Figure 8: I don't doubt the correctness of the figure, but I found it impossible to read. It has 24 panels, all with a _lot_ of information. It took me 5 minutes of staring to even see that there were arrows, which made the last line of the caption make a lot more sense.

We show larger plots of the CESM2-WACCM results as an example and move the old 24 panel figure in the supplementary

[Figure]

**Figure 8.** The squared wavelet coherence between the Nino3.4 (representing ENSO) and the wind-driven ITF transport monthly anomalies under the two SSPs (2015-2100) and two G6 (2020-2100) scenarios in CESM2-WACCM model. The 95% significance level above the background of 1000 Monte-Carlo ensemble of series of identical mean and standard deviation with identical power spectra but phase-randomized Fourier noise (chosen instead of the usual first order autoregressive null hypothesis here because of the strong annual signal; Xia et al. (2023)), is represented by a thick contour line. The arrows indicate the relative phase relationship, that is, in-phase points to the right, anti-phase points to the left, the arrow up indicates that the ITF anomaly leads ENSO by 90°, and a down arrow indicates that the ITF anomaly lags ENSO by 90°. The other models are shown in Fig. S4.

Line 27: Typo in "dimming"

Done.

Lines 101-102: How much smaller?

Changed to "although the average transport was only half the transport observed during INSTANT."

Lines 105-107: This is a bit opaque.

Modified to: While the Amended Island Rule matches the short duration of observed fluxes and variability better than buoyancy, it is possible that changes in buoyancy forcing may affect volume transport of the ITF on decadal scales under a changing climate.

Line 175: How was the interpolation done?

All fields were bi-linearly interpolated (except for sea water vertical velocity, for which we use conservative interpolation) onto a common $0.5° \times 0.5°$ grid.

Line 179: I think you're listing the number of grid boxes, not the resolution.

Yes, thanks.

Line 188: Can you be more specific about which models?

Feng et al. (2011) used an eddy-permitting numerical model, ORCA025, to verify that the Island Rule can capture the decadal variability of the ITF transport.

Line 252: Which test did you use?

Wilcoxon signed rank test.

Line 265: Missing word.

Added "to".

Lines 269-271: I'm not sure I understand this.

Rewritten as : "the equations are the regression trend lines (2015-2100 under the two SSP scenarios and 2020-2100 under the two G6 scenarios) and the significance of the slope".

Line 278: The differences don't look significant (although I'm sure they are). Can you talk more about your significance testing?

Rewritten as "SAI and SD geoengineering methods clearly have different impacts on wind driven contributions to ITF transport for all models (Table S1) and the ensemble mean (Table 2) according to the Wilcoxon signed- rank test, and smaller although still significant differences in upwelling for the 6 model ensemble mean, although significant differences individually only for CESM2-WACCM (Figure 2a, b, Table 2; Table S1)."

Line 282: Here as well - what significance test?

Yes. Table 2 should be below this paragraph which explains the test between scenarios. "all these differences between scenarios are significant ($p<0.05$, Wilcoxon signed-rank test; Table 2)".

Line 293: Can you be more specific about the zonal integration?
We added this explanation to section 3.2: "The contribution of deep ocean upwelling is integrated over the whole Pacific north of 44°S (considering volume conservation and the sill depths of the Indonesian seas is less than 1500 m)."

And this in section 4.1 "zonally integrated, starting at of 44°S and proceeding northward until 60°N, upwelling contributions ".

Corrected.

Yes, the details have been added to Table S1 as in Table 2, which we cite at this point as well in the text.

Line 356: Are these components or different methods? By calling them components, the implication is that you can add them all up and get the "true" ITF.

The wind and upwelling can be summed, the buoyancy is an independent method. The caption is expanded to clarify this: "The differences in monthly ITF Transport (2020-2100)[a] and its components according to the different methods; Wind is the ITF transport derived from Island Rule and used in the Amended Island Rule; Upwelling is the area integral of Pacific upwelling rate at 1500 m used in the Amended Island Rule; Wind and Upwelling is the ITF transport calculated by Amended Island Rule; Buoyancy is the ITF transport by buoyancy forcing and used independently of the other two components."

Line 360: Can you call the "Total" column something different, like "Wind+Upwelling"? Total implies you should be adding in buoyancy.

Yes, replaced as suggested.

Yes, thanks.

Corrected.